# Pulmonary maternal immune activation does not cross the placenta but leads to fetal metabolic adaptation

Signe Schmidt Kjølner Hansen [1,2,3] ✉, Robert Krautz [1,11], Daria Rago[1,2,11], Jesper Havelund[4], Arnaud Stigliani [1,2], Nils J. Færgeman [4], Audrey Prézelin[5,6], Julie Rivière[7,8], Anne Couturier-Tarrade[5,6], Vyacheslav Akimov [4], Blagoy Blagoev [4], Betina Elfving [9], Ditte Neess [4], Ulla Vogel [3], Konstantin Khodosevich [2], Karin Sørig Hougaard [3,10,12] ✉ & Albin Sandelin[1,2,12] ✉

The fetal development of organs and functions is vulnerable to perturbation by maternal inflammation which may increase susceptibility to disorders after birth. Because it is not well understood how the placenta and fetus respond to acute lung- inflammation, we characterize the response to maternal pulmonary lipopolysaccharide exposure across 24 h in maternal and fetal organs using multi-omics, imaging and integrative analyses. Unlike maternal organs, which mount strong inflammatory immune responses, the placenta upregulates immuno-modulatory genes, in particular the IL-6 signaling suppressor *Socs3*. Similarly, we observe no immune response in the fetal liver, which instead displays metabolic changes, including increases in lipids containing docosahexaenoic acid, crucial for fetal brain development. The maternal liver and plasma display similar metabolic alterations, potentially increasing bioavailability of docosahexaenoic acid for the mother and fetus. Thus, our integrated temporal analysis shows that systemic inflammation in the mother leads to a metabolic perturbation in the fetus.

Adaptations to stressors, such as maternal infection and other inflammatory insults, forms a normal part of fetal development[1]. Activation of the maternal immune system, even when caused by transient mild infections, may however also constitute a risk for overwhelming the regulatory capacities of the placenta and thus fetus[1]. Hence, maternal immune activation (MIA) can translate into long-term changes in function and repertoire of responses in offspring organs. In this respect, the fetal nervous system is the most studied[2], but also lasting changes to the immune-[3] and metabolic system[4] are reported. Hence, MIA also interferes with fetal metabolism both acutely and postnatally[4] and may lead to obesity and metabolic diseases in the adult offspring[5,6].

A large body of studies have assessed the acute manifestations of inflammation in maternal and fetal organs induced by LPS, vira or poly I:C administered via intravenous, -peritoneal or -uterine routes[2,4,7]. Much less work has delineated the downstream manifestations of

[1]Department of Biology, University of Copenhagen, Copenhagen, Denmark. [2]Biotech Research and Innovation Centre (BRIC), University of Copenhagen, Copenhagen, Denmark. [3]National Research Centre for the Working Environment, Copenhagen, Denmark. [4]Department of Biochemistry and Molecular Biology, University of Southern Denmark, Odense, Denmark. [5]Université Paris-Saclay, UVSQ, INRAE, BREED, 78350 Jouy-en-Josas, France. [6]Ecole Nationale Vétérinaire d'Alfort, BREED, 94700 Maisons-Alfort, France. [7]Paris-Saclay University, INRAE, AgroParisTech, GABI, 78350 Jouy-en-Josas, France. [8]Paris-Saclay University, INRAE, AgroParisTech, Micalis Institute, 78350 Jouy-en-Josas, France. [9]Translational Neuropsychiatry Unit, Aarhus University, Aarhus, Denmark. [10]Department of Public Health, University of Copenhagen, Copenhagen, Denmark. [11]These authors contributed equally: Robert Krautz, Daria Rago. [12]These authors jointly supervised this work: Karin Sørig Hougaard, Albin Sandelin. ✉e-mail: signe.sk.hansen@sund.ku.dk; ksh@nfa.dk; albin@bio.ku.dk

immunogens targeted to the lungs, including maternal airway inflammation arising from environmental dust and respiratory infections[8–14]. The latter are common during pregnancy: 49.6% of control mothers in the National Birth Defects Prevention Study reported respiratory infections during pregnancy[15,16]. A considerable part of community-acquired pneumonia owes to infections with Gram-negative bacteria[17]. It is therefore surprising that in a recent systematic review of 118 MIA studies, no studies administered LPS via the airways[7].

A limitation of most MIA studies is that they measure only single responsible mediators and/or capture the response in single organs at a single timepoint. Investigation of fetal effects of maternal LPS lung administration requires profiling of several tissues and timepoints[18]. First, profiling the maternal lung is relevant as it is the onset of inflammation: this will show the strength and duration of the directly induced inflammation. Second, it is important to study the degree to which the lung inflammation translates into a systemic immune response, its duration, and whether inflammatory messengers from the lung, such as cytokines, are exported to the maternal bloodstream. Relevant organs to profile to answer these questions include the maternal liver and plasma. Third, to understand if and how maternal states affect the fetus, the placenta is important to consider. The placenta has a remarkable range of physiological functions that are pivotal for pregnancy homeostasis and fetal development[19]. However, its rich vasculature also makes it receptive to maternal blood-borne pathogens and inflammatory mediators[20]. The cell layers separating maternal and fetal circulation therefore have specialized and potent defense/barrier mechanisms, such as morphological plasticity[21] and maintenance of endothelial integrity[22]. In the context of MIA, regulated placental immune responses are crucial in order to limit the transfer of inflammation from mother to fetus[4]. Fourth, to understand the nature and duration of effects in the fetus, one or more fetal organs must be profiled. To study effects on a shorter time span, the fetal liver is particularly important because it receives most of the fetal blood that has passed the maternal-fetal interface and because its metabolic functions have the potential to acutely affect the homeostasis of the whole fetus[23].

To this end, we here characterize the temporal response to maternal lung inflammation across the maternal and fetal organs mentioned above, using multi-omics, imaging, and integrative analyses. We show that maternal organs responded strongly, activating innate immune response genes, whereas the placenta did not: instead, it orchestrated a specific adaptive response, comprising upregulation of immune modulatory and tissue-integrity genes and downregulation of cell growth genes. The fetal liver did not upregulate immune response genes but carried out metabolic adaptations, including an increase in the proportion of lipids containing essential fatty acids (EFA) and docosahexaenoic acid (DHA) which may lead to increased bioavailability for the crucial development of fetal organs such as the brain. The maternal liver and plasma displayed the same lipid response pattern, although at earlier timepoints than in the fetal liver, thus suggesting that maternal DHA may become available to the fetus within the studied timeframe.

## Results

### Acute pulmonary LPS response does not extend to the placenta

To investigate gene expression response following MIA, we exposed pregnant C57BL/6 mice (gestation day (GD) 17) to 1 μg of lipopolysaccharide (LPS) or $H_2O$ vehicle (Ctrl) by intratracheal instillation. This dose was chosen to model a robust airway inflammation without causing excessive lung injury or preterm birth (see Supplementary Note 1 and Supplementary Fig. 1A, B for dose choice considerations).

Mice were sacrificed 2, 5, 12, or 24 h after instillation, and maternal lung and liver, placenta, and fetal liver were excised (Fig. 1A). Fetal tail DNA was genotyped for sex and only one female pup per dam was used. The placenta (chorionic plate, labyrinth, and junctional zones) and decidua were separated manually (Fig. 1A). Of note, at GD17 the

labyrinthine structure of maternal blood spaces and fetal vessels[24] hampers full anatomical separation and these samples should therefore be regarded as decidua- and placenta-enriched. RNA was extracted from 7–10 biological replicates for each combination of treatment, timepoint, and tissue (total 370 samples, from 74 dam-fetus pairs). In the lung, serum amyloid A (SAA) proteins correlate closely with neutrophil influx, a hallmark of lung inflammation[25,26]. In this study, mRNA levels of *Saa3*, measured by qPCR, were highly increased throughout the 24 h in LPS compared to Ctrl maternal lungs (Supplementary Fig. 1C). Furthermore, LPS-induced maternal weight loss, leveling at 12 h with 6–8% decrease vs. time-matched Ctrl (Supplementary Fig. 1D). Overall, these observations are consistent with LPS inducing strong inflammation[27,28].

RNA samples were subjected to paired-end, polyA-selected RNA-seq. Reads were mapped to the mouse transcriptome (M23); all 370 libraries were retained after quality control. Principal component analysis showed that samples clustered by tissue and timepoint (Supplementary Fig. 1E, F). We calculated the average LPS vs. Ctrl $\log_2$ expression fold change ($\log_2$FC) for each tissue and timepoint and visualized genes with an absolute $\log_2$FC > 1 in at least one tissue and timepoint as a heatmap (Fig. 1B). This enabled four important observations, also confirmed by differential expression (DE) analysis (Supplementary Fig. 1G):

First, the maternal lung showed strong transcriptional responses at 2–5 h, partially persisting at 12–24 h. Gene ontology (GO) analysis showed that upregulated biological processes were dominated by pathways related to acute-phase signaling, including response to lipopolysaccharide (e.g. the TLR4 pathway), cytokine production, chemotaxis and fever generation (Fig. 1B, Supplementary Fig. 2A). This corresponds well with the molecular dynamics described previously in mice following LPS airway challenge[27,28]. Together with the observed maternal weight loss, these observations are consistent with LPS inducing strong inflammation.

Second, also the maternal liver responded to LPS, albeit less strongly. The main response occurred at 2–5 h, and many upregulated genes were shared with the lung. Upregulated genes related to activation of the innate immune response and inflammatory pathways. The overlap with observations in mice injected with LPS[29] indicates a direct response to LPS[29] (Fig. 1B, Supplementary Fig. 2B). Importantly, GO analysis also showed metabolic processes such as lipid modification and catabolism of fatty acids to be enriched for upregulated genes at 12–24 h, while biosynthesis of long-chain fatty acids was dominated by downregulated genes at 24 h (Supplementary Fig. 2B). This indicates a simultaneous indirect hepatic response to LPS, which will be explored further.

Third, the placenta and decidua also responded to LPS, but qualitatively differently from the maternal lung and liver. Genes responded mainly at a single timepoint: the largest change occurred at 5 h (Fig. 1B). Notably, decidual and placental LPS vs. Ctrl responses were overall similar (Fig. 1B, Supplementary Fig. 2C). This may reflect similar response patterns, or the difficulty in separating tissues of maternal and fetal origin, as discussed above. Therefore, when analyzing expression response to LPS (e.g. by LPS vs. Ctrl $\log_2$FC), we combined placental and decidual samples, and denoted this 'placenta+decidua'. For clarity, when analyzing gene expression levels in respective tissues rather than LPS response (e.g. transcripts per million (TPM)), we use 'decidua' and 'placenta' in the text.

Fourth, the fetal liver displayed a unique response to LPS compared to other tissues, but with a magnitude similar to that of the placenta in terms of number of DE genes (Supplementary Fig. 1G) and range of $\log_2$FC values. As in the placenta, the largest number of DE genes was observed at 5 h.

Thus, there was a stark contrast between the strong and long-lasting LPS response in the maternal lung and liver and the fainter and temporally more restricted response in the decidua, placenta, and fetal

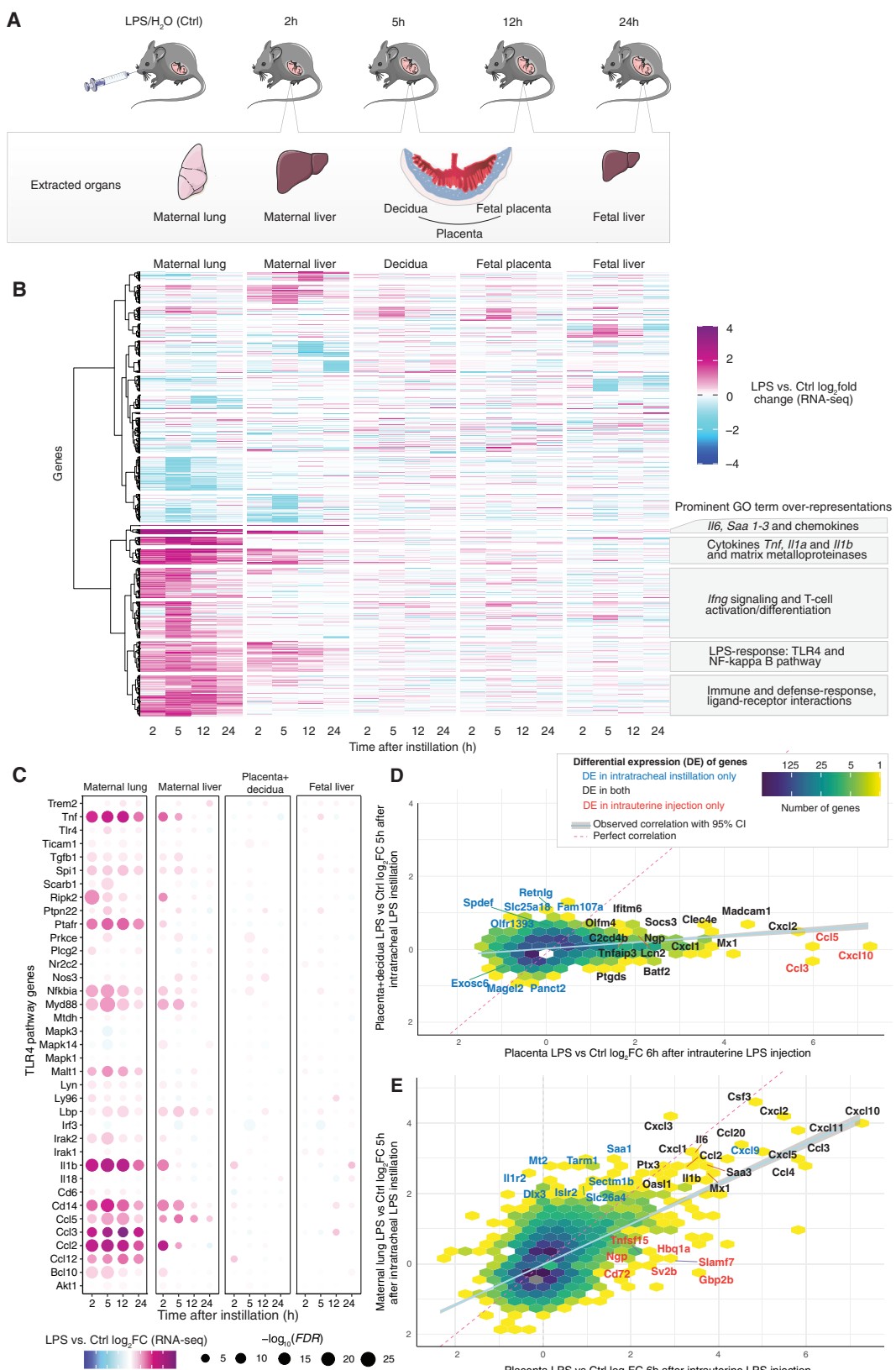

liver. As the placenta regulates maternal-fetal signaling and resource allocation[30], its response is key to understanding placental and fetal strategies relative to maternal systemic inflammation. Thus, we profiled the expression of selected key genes from the LPS-TLR4-signaling pathway across organs as a proxy for direct response to LPS and a resulting immune response.

Although placental and decidual cells can induce the TLR4 pathway upon LPS stimulation[31–35], we observed only minor upregulation of genes in this pathway in placenta+decidua, at any timepoint (Fig. 1C). We, therefore, speculated whether the placenta responded indirectly, e.g. to inflammatory messengers arising from the inflamed maternal lung, rather than directly to LPS itself. To compare this presumably

**Fig. 1 | Experimental design and RNA-seq overview. A** Experimental design. At gestational day 17, mice were intratracheally instilled with lipopolysaccharide (LPS) or vehicle ($H_2O$, denoted Ctrl). At set timepoints after instillation, mice ($N = 7$–9 for each group/timepoint) were sacrificed and organs were dissected (gray box) from mothers and one female fetus and used for subsequent analyses, including RNA-seq (**B**). Plasma samples were also collected from mothers at all timepoints. Artwork adapted from bioicons (https://bioicons.com/, CC 0 license). **B** Overview of RNA-seq results. Rows show genes that changed expression ≥2-fold in at least one timepoint and tissue. Columns show time after LPS or Ctrl instillation, sorted first by tissue and then by time. Colors indicate LPS vs. Ctrl gene expression $\log_2$ fold change ($\log_2$FC), for respective timepoint and tissue. Callouts show major gene ontology term enrichments for subclusters dominated by maternal lung and liver. **C** Expression change of TLR4 pathway genes. Heatmap organized as in (**B**), but uses placenta+decidua samples for differential expression (DE) analysis and shows expression change (placenta+decidua $\log_2$FC, indicated by color) and significance

($-\log_{10}$ FDR, indicated by dot size). **D** Comparison of placental gene expression change following intratracheal instillation vs. intrauterine LPS injection. X-axis shows placental gene LPS vs. Ctrl 6 h gene expression $\log_2$FC after intrauterine LPS injection (data from ref. [36]). The Y-axis shows placenta+decidua LPS vs. Ctrl gene expression $\log_2$FC 5 h after intratracheal LPS instillation (our data). Hexagon colors indicate the number of genes. Callouts show DE genes, colored by whether the gene was DE in one or both experiments. The dotted line indicates X = Y: blue line indicates the observed correlation with 95% gray confidence interval. **E** Comparison of maternal lung gene expression change following intratracheal LPS instillation vs. placental gene expression following intrauterine LPS injection. Organized as in D, but Y-axis shows maternal lung LPS vs. Ctrl gene expression $\log_2$FC 5 h after intratracheal LPS instillation (our data). Source data in fig1_response_matrix_list.rds, fig1_limma_results_no_maternal_contrasts.csv and fig1_lien_fold_change_summary.rds.

indirect response with a placental response to intrauterine LPS injection, we compared our 5 h placenta+decidua RNA-seq data to RNA-seq data from a mouse study using intrauterine LPS injection[36]. The datasets were comparable in terms of gestational stage (GD17.5 and GD17) and timepoint after exposure (5 h and 6 h), although the intrauterine LPS dose (50 μg) was designed to induce preterm birth (appr. 12 h after exposure) and was considerably higher than ours (1 μg). The placenta+decidua responses to intratracheal LPS instillation (our data) and intrauterine LPS injection[36] correlated moderately: a small set of genes were upregulated in both experiments, including antimicrobial and immune modulators e.g. *Socs3, Clec4e, Batf2, Lcn2, Madcam1, Olfm4,* and *Mx1* (Fig. 1D). Intratracheal LPS instillation, but not intrauterine LPS injection, induced upregulation of a diverse set of genes including actin-dynamics modulator *Fam107a,* and immunomodulator *Retnlg.* Conversely, only intrauterine LPS injection upregulated the chemokines *Ccl3, Ccl5, and Cxcl10* (Fig. 1D). Irrespective of the difference in dose, the placental expression changes following intrauterine LPS injection at 6 h were highly correlated to those of the LPS-instilled lungs at 5 h in our study (Fig. 1E), and GO analysis showed similar roles for upregulated genes in the two tissues (Supplementary Fig. 2D). Of note, GO terms associated with regulation of immune response were highly enriched in placentas subject to intrauterine LPS injection and not LPS-instilled lungs.

We conclude that although the placenta has the capacity to mount a strong direct LPS response[34,36,37], possibly enhanced by a direct, high LPS exposure, it reacts fundamentally differently to LPS lung administration, characterized by immune modulation rather than activation.

## IL-6 may induce placental immunomodulation via SOCS3
We hypothesized that the specificity and temporality of placental gene expression response were contingent on one or more circulating cytokines/chemokines secreted by the maternal lungs in response to LPS[38]. Therefore, we measured concentrations of 11 cyto- and chemokines (CCL2, 4, 7, 11, 17, 20, 24, CXCL1, 13, 16, and IL-6) in maternal plasma at all timepoints using the Luminex assay (Supplementary Fig. 3A). We compared maternal plasma protein abundance changes (LPS vs. Ctrl $\log_2$FC by Luminex) with expression changes of the corresponding mRNAs in maternal lung (LPS vs. Ctrl $\log_2$FC by RNA-seq). Generally, cyto/chemokine concentrations in maternal plasma were increased at 2 h but decreased to Ctrl levels at 24 h (Fig. 2A and Supplementary Fig. 3B). Corresponding lung mRNA levels were generally increased at 2 h but decreased slower than protein levels. Notably, *Il6* and *Ccl20* mRNAs were highly upregulated in LPS-instilled lungs, especially at 2, 5 and 12 h ($\log_2$FC = 2.8–3.5). Their protein levels in maternal plasma were highly increased at 2 h ($\log_2$FC > 1.5) and decreased gradually to Ctrl levels at 24 h (Fig. 2A, left). The levels of other measured chemokines fell into two categories: (i) CCL7, CCL11, and CXCL13 were 2-fold upregulated in plasma at 2 h (Fig. 2A, right) but were not highly induced at mRNA levels in lung, and (ii) CXCL1, CCL2,

CCL4, CCL17, CCL16, and CCL24 were only modestly upregulated in plasma ($\log_2$FC < 1 at all timepoints), and only *Cxcl1, Ccl2, and Ccl4* mRNAs were highly upregulated in lung (Supplementary Fig. 3B). Therefore, we reasoned that IL-6 and CCL20 were the most prominent candidates for mediating systemic responses while CCL7, CCL11, and CXCL13 might mediate local inflammatory responses. Because the liver contributes to combating systemic inflammation, it might constitute an additional source of cyto/chemokines[39]. Therefore, we also compared the upregulation of cyto/chemokines in maternal plasma with mRNA upregulation in the maternal liver. Surprisingly, neither *Il6* nor *Ccl20* gene expression was upregulated by the maternal liver (Fig. 2B, left), while *Ccl7* and *Cxcl13* gene expression was moderately increased (Fig. 2B, right). Among the remaining measured cytokines which were not highly upregulated in plasma, only *Ccl2* and *Cxcl1* mRNAs were highly upregulated in the liver (Supplementary Fig. 3C). Therefore, while the maternal liver might contribute to the temporary plasma level increase in CCL7 and CXCL, the prominent increase in IL-6 and CCL20 is likely due to their high mRNA upregulation in the lungs.

We reasoned that the chemo/cytokines most likely to induce placental gene expression would have increased levels in maternal plasma and highly expressed cognate receptor(s) in decidua and placenta. Hence, we correlated changes in IL-6 and CCL20 plasma concentrations (Luminex $\log_2$FC, LPS vs. Ctrl) with the mRNA expression levels (RNA-seq TPM) of their receptor(s) in placenta and decidua, based on CellPhoneDB[40] and manual curation.

Although CCL20 abundance in maternal plasma was increased following LPS instillation (Fig. 2A, left), its receptor *Ccr6* was not expressed at any timepoint in decidua nor placenta regardless of treatment (Fig. 2C). Therefore, the effect of CCL20 on placenta is likely only minor, similarly to the rest of the measured cytokines (Supplementary Fig. 3B, C and Supplementary Note 2).

Conversely, IL-6 levels were highly increased in maternal plasma, and its two receptor-subunit genes, *Il6ra* and *Il6st*, were constitutively expressed in decidua and placenta regardless of treatment (Fig. 2D), consistent with the well-known role if IL-6 as a mediator of fetal effects in models of MIA[41–44] induced by bacteria[41–44] and poly I:C[41–44]. Interestingly, decidua had significantly higher gene expression levels of *Il6ra* at all timepoints regardless of treatment, and significantly higher expression levels of *Il6st* at 2 h in both treatments, and at 5 h in LPS-treated mice (Fig. 2E, P < 0.05, two-sided Mann–Whitney tests). In classical IL-6 signaling, IL-6 binds to IL6RA with relatively low affinity, followed by recruitment of IL6ST to form a high-affinity complex[45]. Since the median *Il6ra* gene expression was >2.5 times higher than that of *Il6st* in both tissues (Fig. 2E), changes in *Il6ra* abundance will be rate limiting, and thus suggests that the decidua might be more receptive to IL-6 receptor binding than placenta (Fig. 2E). We reasoned that while the IL-6 pathway may be activated in both tissues following LPS instillation, the gene expression response of IL-6 pathway genes may be different in decidua and placenta. We therefore visualized

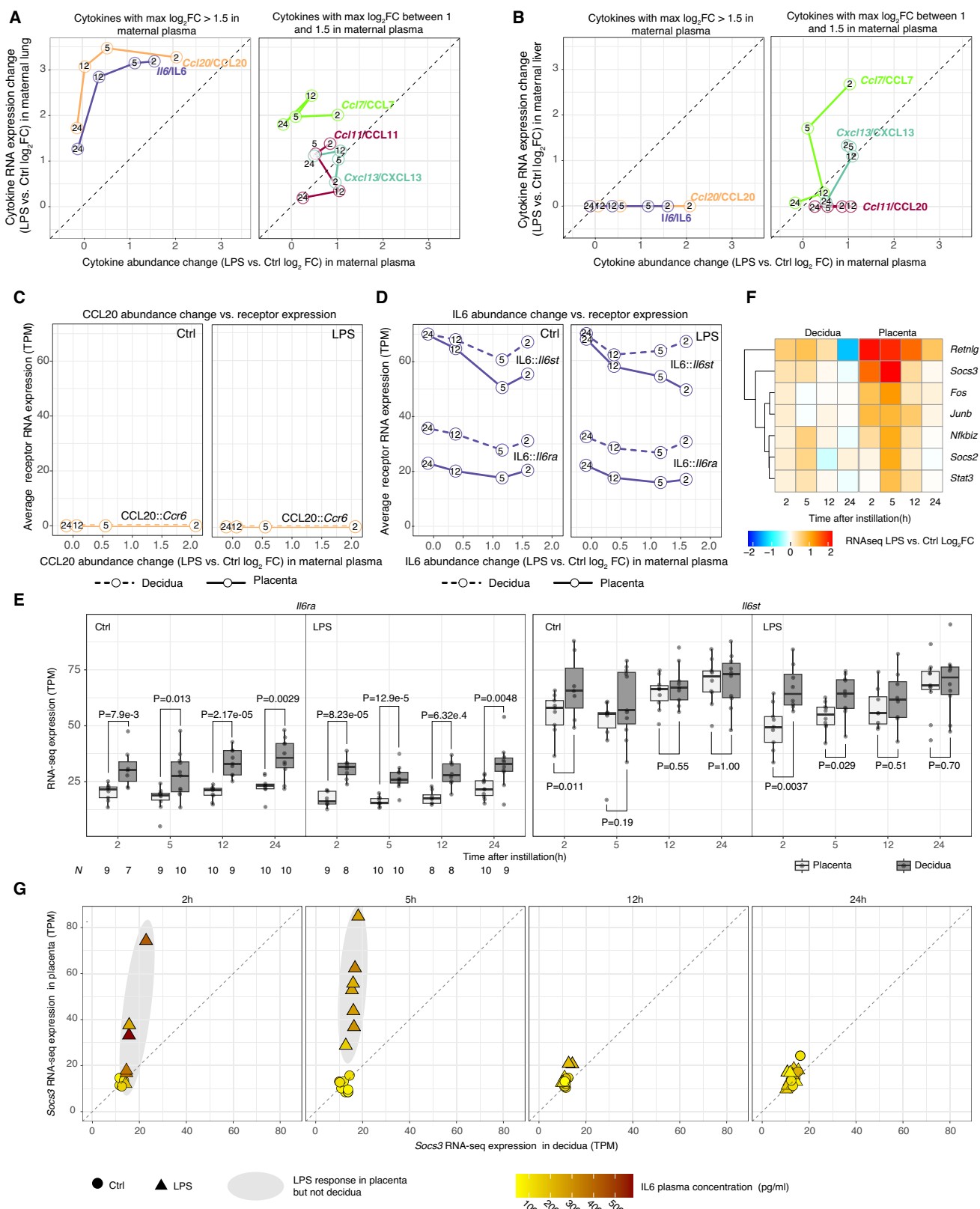

expression changes (log₂FC LPS vs. Ctrl) over time in the decidua and placenta of genes associated with the Jak/STAT/IL-6 signaling pathway as a heatmap (Fig. 2F shows the most changing genes). Strikingly, key inflammation regulators *Stat3, Socs2, Socs3,* and *Nfkbiz* were upregulated exclusively in the placenta 2–12 h after LPS exposure. The transcription factor STAT3 regulates activation of Jak/STAT/IL-6-pathway targets while SOCS2- and SOCS3 inhibit STAT3 and thereby IL-6-induced inflammation[46,47]. *Nfkbiz* can also be induced by STAT3 and inhibits pro-inflammatory signaling by NFKB and TNF[48,49]. The gene expression of *Junb* and *Fos*, regulatory targets of IL-6-signaling[50], were also upregulated in the placenta at 2–12 h. Thus, while mRNAs for IL6RA and IL6ST proteins that are necessary for inducing IL-6 signaling

**Fig. 2 | Blood cytokine-receptor signaling and downstream effects. A** Maternal lung cytokine mRNA expression vs. maternal plasma protein levels. Y-axis shows maternal lung mRNA lipopolysaccharide (LPS) vs. Ctrl log$_2$ fold change (log$_2$FC). X-axis shows maternal plasma LPS vs. Ctrl log$_2$FC of corresponding proteins. Cytokines are indicated by color: numbers indicate hours after instillation. Dotted lines show X = Y. **B** Maternal liver cytokine mRNA expression vs. maternal plasma protein levels. As in (**A**), but Y-axis shows maternal liver mRNA expression change. **C** Maternal plasma CCL20 abundance change vs. mRNA expression of *Ccr6* in decidua and placenta. Y-axis shows maternal plasma CCL20 LPS vs. Ctrl log$_2$FC. X-axis shows the average *Ccr6* mRNA expression level in Ctrl decidua (dotted lines) or placenta (solid lines). Line colors and numbers as in (**A**). Subpanels show Ctrl and LPS treatment. **D** Maternal plasma IL-6 abundance change vs. decidua and placenta mRNA expression of *Il6ra* and *Il6st*. As in (**C**), but shows IL-6::Il6ra and IL-6::Il6st pairs. **E** mRNA expression of *Il6ra* and *Il6st* in decidua and placenta. Y-axis shows mRNA expression in decidua (gray) and placenta (white). Dots show samples and X-axis hours after instillation. *T*wo-sided Mann–Whitney test *P*-values between distributions are shown, bold when *P* < 0.05. Subplots show Ctrl and LPS-treated mice. Receptor name on top. Boxplot center = median, box = interquartile range, whiskers = max 1.5*interquartile range from hinge; all data points shown. Sample sizes for each group/timepoint are shown in Il6ra subplots (same for Il6st). **F** Heatmap of the most LPS instillation-responding genes in the IL-6 pathway. Rows show genes. Columns show hours after instillation. Tissue on top. Cell color indicates LPS vs. Ctrl RNA-seq log$_2$FC. **G** *Socs3* gene expression change in paired decidua and placenta samples. Y and X-axes show *Socs3* gene expression in the placenta and decidua. Dots show dam-fetus pairs. Dot shape indicates treatment. Color shows maternal plasma IL-6 concentration. Dotted lines indicate X = Y. Highlights show *Socs3* upregulation in LPS-treated placenta. Source data in fig2_cytokines_conc.csv, fig1_limma_results_no_maternal_contrasts.csv and fig2_tpm_tibble.rds.

had higher expression levels in decidua than placenta regardless of exposure, the genes of the key IL-6 pathway inhibitors *Socs2* and *Socs3* were only upregulated in the placenta following LPS treatment. We hypothesized that IL-6 signaling is inhibited by increased *Socs3* gene expression, and therefore plotted *Socs3* gene expression in decidua vs. placenta for each mouse, where dots were colored by paired circulating IL-6 levels (Fig. 2G). This showed that (i) increase of *Socs3* expression in the placenta following LPS exposure (triangles in Fig. 2E) at 2–5 h was observed in all but one LPS-treated mouse, while no mouse increased expression of *Socs3* in the decidua regardless of LPS treatment and (ii) all mice whose *Socs3* expression increased at 2 or 5 h also had higher circulating IL-6 levels at the same timepoint.

To validate this observation on protein level, we imaged SOCS3 abundance and localization with immunohistochemistry on midline-sections of whole placentas from Ctrl and LPS samples in the 5 h group (Fig. 3). Samples from the LPS group displayed high SOCS3 staining in spongiotrophoblast cytoplasm across the junctional zone (Fig. 3A–C). SOCS3 expression was also present in the labyrinth zone cytoplasm of LPS samples, although less intensely. In contrast, Ctrl samples at 5 h displayed only faint SOCS3 cytoplasmic staining in the junctional zone (Fig. 3D–G). These staining-patterns are consistent with a previous IHC-based study showing increased SOCS3 expression in the junctional zone[51] of hyperglycemic rat placentas.

As a summary, our data suggests that IL-6 from the maternal lung is a main candidate for evoking a response in the decidua and placenta, characterized by regulation of IL-6-signaling and inhibition by SOCS3 in the placenta. The decidua increases IL-6 receptor expression over time, while the placenta temporarily increases the expression of inflammation-inhibitory SOCS3 in the junctional zone. These fine-tuned temporal dynamics propose a mechanism by which particularly the placenta and to a lesser degree the decidua can abolish the propagation of an inflammatory response and instead maintain an immuno-modulatory profile.

## Placental response to pulmonary LPS instillation

Since the placenta reacted differently to LPS instillation than maternal lung and liver we characterized the placenta+decidua gene expression response in detail. Across all timepoints, 484 genes were DE (FDR < 0.05, |log$_2$FC| >0.5). The largest number of DE genes were found at 5 h (313 genes; Supplementary Fig. 1G). GO analysis showed distinct functional annotation enrichments for each timepoint, except 2 h (see source data file GO_enrichment_results_RNAseq.zip). We grouped functionally related GO terms into 'GO themes' (Fig. 4A–F, left). For each GO theme, we selected a subset of genes that were among the 50 most DE at the timepoint of GO theme enrichment and visualized their LPS vs. Ctrl expression change over time as heat maps (Fig. 4A–F, middle).

At 5 h, two GO themes were enriched in upregulated genes: (i) 'Cell adhesion', including spreading and migration, and actin filament-based process terms, and (ii) 'Cell cycle delay', including small GTPase- and Ras signal transduction, and negative regulation of cell population proliferation terms (Fig. 4A, B, left). Because Ras-signaling regulates cytoskeletal dynamics associated with cell adhesion and migration[52], both themes may denote tissue-integrity functions. We therefore analyzed genes within these themes jointly. Upregulated genes were involved in cytoskeleton-mediated strengthening of tissue integrity at different levels, including Rac signaling and cell migration (e.g. *Csf1, Camk2d, Arhgef3, Cdc42, Dock2*), Ras/RhoA signaling and adhesion (*Cavin4, Ctnnal1, Gnai13, Kalrn, Kras*) and Rab/endosomes/endocytosis (*Agtr1a, Dynlt1c, Arfgef2, Cyth4, Dennd3*; Fig. 4A, B, middle panel).

Two GO themes were enriched in downregulated genes at 5 h. One encompassed 'Mitochondria' (Fig. 4C, left), including mitochondrial assembly (e.g., *Ndufaf2, Timm21, Polrmt, Dmac2, Rmnd1*), mitochondrial ribosome formation (e.g. *Mrpl34, Mrpl39, Noa1*) and sensing of mitochondrial stress (e.g., *Oma1, Prkaa1, Tufm, Gadd45gip1*; Fig. 4C, middle). Second, genes associated with GO terms related to 'RNA processing' were enriched in downregulated genes at 5 h (Supplementary Fig. 4A). In addition, 'Ribosomes and translation'-related GO terms contained some genes that were downregulated at 5 h (see below).

Based on the downregulation of mitochondrial production, RNA processing, ribosomes, and translation genes, we hypothesized that at 5 h, cell growth is reduced. We suggest that by strengthening cell adhesion in endothelial and epithelial cell layers and reducing changes in architecture via proliferation, the placenta restricts maternal-fetal transport of putative infectious agents[21,53,54] (Fig. 4A–C, right). Such adverse signals may comprise diffusion- or receptor-mediated transport of inflammatory mediators across the placental epithelium, immune cell infiltration, or translocation of microbial infectious organisms. These changes in cell architecture would largely be mediated by Ras superfamily genes, which were upregulated at 5 h. These mechanisms appeared to gradually decrease, as only some genes upregulated at 5 h remained so at 12 h (Fig. 4A–C, middle).

At 12 h, GO terms associated with 'Glycosylation and mannosylation' were enriched in downregulated genes (Fig. 4D, left). One group of genes was part of alpha-dystroglycan (DAG1) signaling, either assisting in binding or glycosylation of DAG1 (*Large1, Pomgnt1-2, Pomt1-2, Fktn, Fkrp*) or as part of Notch signaling (*Pofut1, Poglut1*) which may be tuned by DAG1[55]. The dystroglycan complex links the extracellular matrix with the intracellular cytoskeleton, is regulated by glycosylation, and is associated with tissue remodeling/structural changes in the placenta[56]. We speculated that reduced glycosylation of DAG1 can lead to reduced cell-matrix adhesion, ECM stiffness, and tissue integrity, similar to glycosylation roles in adhesion, cell communication, and infection of endothelial cells[57,58] (Fig. 4D, right). This could represent a reversal of the increased adhesion and tissue integrity at 5 h and correlates with the decrease in maternal plasma cytokine concentrations at 12 h (Fig. 2A). It implies the reestablishment

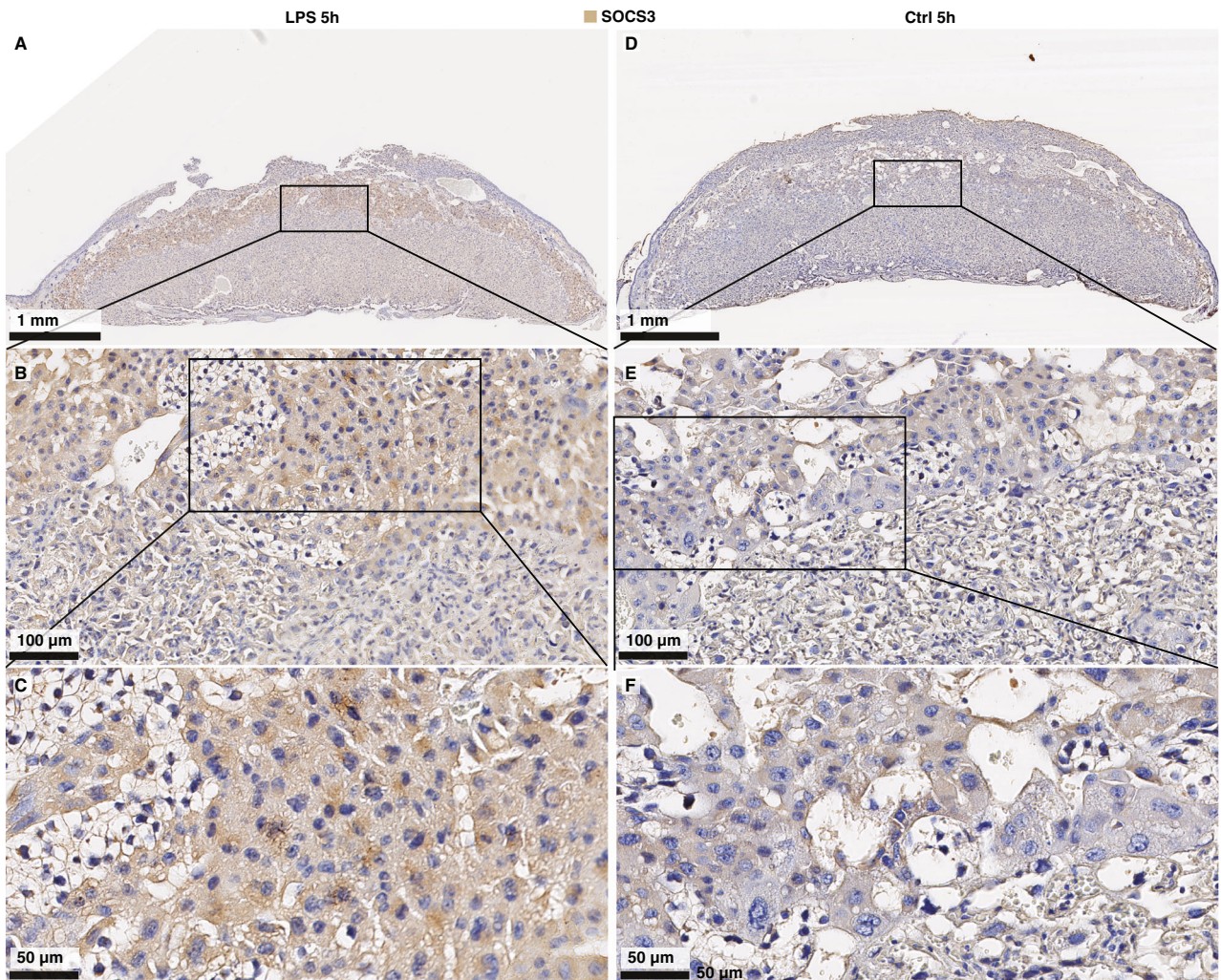

**Fig. 3 | Imaging of SOCS3 protein expression in the placenta. A–F** Representative images of SOCS3 expression in the whole placenta at 5 h after Ctrl/lipopoly-saccharide (LPS) instillation. Nuclei are marked by purple (diaminobenzidine (DAB) staining), SOCS3 by brown-orange. Top row (**A, D**) shows overview images (×3.4 zoom) of the whole placenta from LPS- (left) and Ctrl mice (right). The border of part of the junctional zone (JZ) is indicated by dotted lines. Second and third rows show 20× and 40× zoom-ins of selected regions, indicated by rectangles, containing parts of the labyrinth- and junctional zones. Size bars are shown in each image. Immunohistochemistry was repeated three times, each time with *N* = 3–5 for each condition. Source data in folder fig3_SOCS3_imaging.

of homeostatic cell-ECM interactions and adhesion, possibly due to a decrease in sensing of infectious threats.

One GO theme pattern encompassed more than one timepoint: genes associated with 'Ribosomes and translation' terms were partially downregulated at 5 h and highly upregulated at 24 h (Fig. 4E). At 12 h, some of these genes remained downregulated while others were upregulated, suggesting a shared trend with different time trajectories. Genes following this pattern were involved in rRNA processing and ribosome biogenesis (*Nol9, Utp20, Npm3*), ribonucleoproteins (*Rrp9, Gemin8, Ddx20*), translation/protein synthesis (*Wdr55, Pus7, Eif4b, Eif2a, Eif2b3*) and transcriptional regulation (*Noc2l, Per2, Pelp1, Sarnp*) (Fig. 4E, middle).

The increase in ribosome biogenesis and protein synthesis at 12–24 h was accompanied by a marked upregulation of genes involved in endoplasmic reticulum (ER) stress (Fig. 4F), including genes related to ER-associated protein degradation (ERAD; *Derl2, Syvn1, Herpud1, Vcp, Erlin1*), unfolded protein response (UPR; *Dnajc3, Dnajb9, Ficd*), ER-chaperones (*Hspa5, Hsp90b1, Sdf2l1*) and ER stress marker genes (*Pik3r2, Uba5, Preb, Cdk5rap3*) (Fig. 4F, middle).

The patterns of the two themes above show the placenta's capability of adaptation: the increased ribosomal, RNA processing, and protein synthesis activity at 12–24 h may be a compensation for the decrease of the same processes at 5 h. The increase in protein biosynthesis and RNA metabolism at 12–24 h may be linked to the increase in ERAD/UPR signaling and chaperone levels during the accelerated protein production in the ER lumen needed for returning to homeostasis (Fig. 4E, F, right panel).

The unfolded protein response (UPR) induced during ER stress can be divided into three signaling pathways, where the PERK pathway activated by the transcription factor ATF4 initiates the mildest response, restoring ER function by attenuating non-essential protein synthesis[59]. We reasoned that ATF4 expression and localization in the placenta would indicate ER stress, since ATF4 translocation to the nucleus marks the activation of the PERK pathway[60]. Immunohistochemistry of placenta sections stained with ATF4 at 24 h showed a distinct and frequent expression in cell nuclei of syncytiotrophoblasts throughout the labyrinthine zone of the LPS group (Fig. 5, ATF4 staining visible as red dots), while nuclear localization was observed in only a few cells in placenta sections from the Ctrl group.

### Analysis of placental protein phosphorylation changes

Protein phosphorylation is essential for adjustments to environmental changes[61,62]. We reasoned that part of the earliest placental response might include phosphorylation cascades and/or phosphorylation

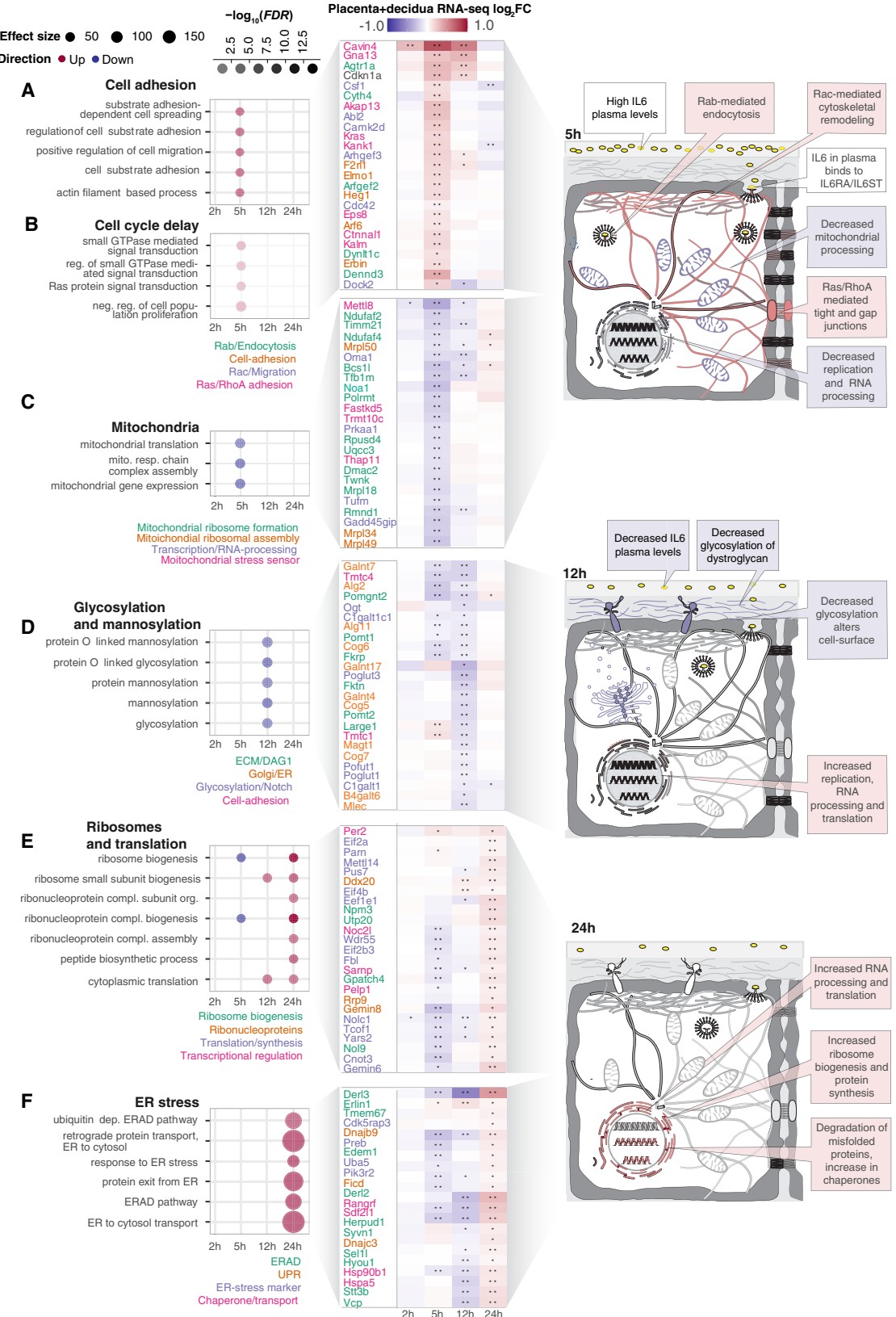

changes accompanying protein abundance changes following the observed gene expression patterns.

We measured LPS-induced phosphosite changes at 2 and 5 h in placenta+decidua samples ($n = 7–9$ for each group/timepoint) using LC-MS/MS phosphoproteomics. Few changes in phosphorylation sites occurred at 2 h (216 sites in 98 proteins, $FDR < 0.05$) while numbers were higher at 5 h (882 sites in 578 proteins; Supplementary Fig. 5A, B). GO analysis showed that changes primarily occurred in proteins mirroring the functional roles of genes upregulated at 5 h (Fig. 6A and Supplementary Fig. 5A); in particular, proteins associated with DNA metabolism, transcriptional regulation, cell growth, and RNA/protein processing or metabolism (Fig. 6A). We also observed over-

**Fig. 4 | Detailed analysis of placental gene expression change. A** Over-representation of adhesion-associated gene ontology (GO) terms and the expression change of key associated genes across time in placenta+decidua. Left panel rows show selected adhesion-associated GO terms (cell adhesion GO theme) and their over-representation in up (red) or downregulated genes (blue) for a given timepoint after instillation in placenta+decidua (X-axis). Dot size and color intensity show effect size and statistical significance ($-\log_{10}(FDR)$). Middle panel shows placenta+decidua LPS vs. Ctrl $\log_2$FC expression change of top differentially expressed (DE) genes linked to one or more of these GO terms as a heatmap where rows indicate genes, columns indicate time after instillation and cell color indicates lipopolysaccharide (LPS) vs. Ctrl $\log_2$FC. Stars indicate significant DE (LPS vs. Ctrl,

$FDR < 0.05$). Gene names are shown to the left, colored by functional roles. Right cartoon summarizes expression changes: cell structures or organelles whose genes are up- or downregulated as a response to LPS instillation are colored red and blue, respectively. The adhesion and cell cycle delay GO themes (panel B) have similar dynamics, therefore, genes (middle panel) are shown from both themes. Gray backgrounds link GO themes, genes, and cartoon summary (right). **B–F** GO term over-representation and expression of linked genes for different GO themes. Plots are organized as in A, but show different GO themes, GO terms, and their associated genes. Source data in fig1_limma_results_no_maternal_contrasts.csv and GO-analysis data in folder GO_enrichment_results_RNAseq. Artwork adapted from bioicons (https://bioicons.com/, CC 0 license).

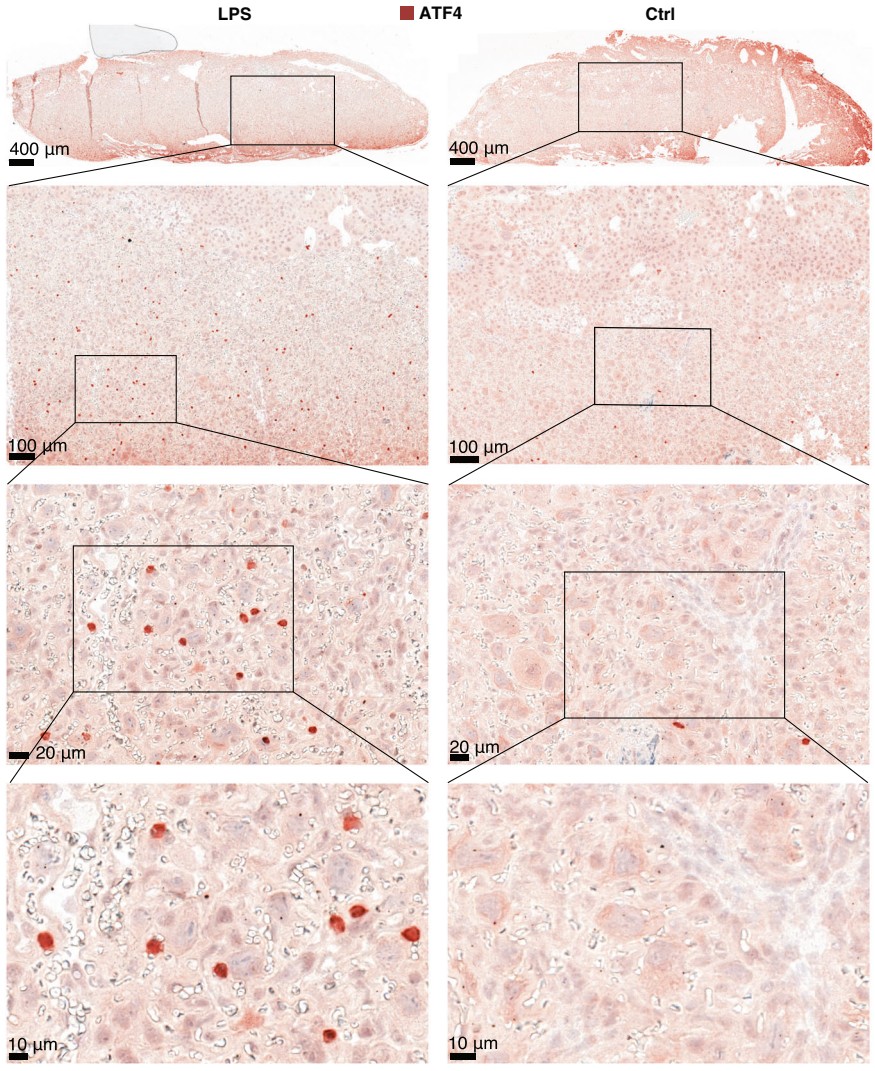

**Fig. 5 | Imaging of ATF4 protein expression in the placenta.** Representative images of ATF4 localization in the labyrinth zone of placenta at 5 h after lipopolysaccharide (LPS) or Ctrl instillation (left and right column, respectively). Nuclei are marked by purple, ATF4 by red. Rows display increasing magnification levels (1.36, 5.6, 20 and 40×). Size bars are shown in each image. Immunohistochemistry repeated twice, each time with $N = 4–5$ for each condition. Source data in fig5_ATF-4_imaging.zip.

representation of damage response terms such as viral processes and DNA damage. Phosphorylation changes in DNA damage-associated proteins may imply regulation of growth processes.

Interestingly, several phosphoproteins had increased phosphorylation at 2 h but decreased phosphorylation at 5 h, comparing LPS and Ctrl (Fig. 6B, left). Notably, many of these proteins have roles in tissue integrity, RNA processing and chromatin remodeling (Fig. 6B, left); themes that were also enriched in upregulated genes at 5 h on RNA

level (Fig. 6, Supplementary Fig. 5A). Despite this similarity, proteins with these phosphorylation patterns were not DE at mRNA level (Fig. 6B, right). This suggests the existence of an early protein phosphorylation wave which affects other proteins than those being transcriptionally upregulated. Phosphorylation of several proteins with this temporal pattern (e.g. ATRX, HMGA2, NUDC) is cell cycle dependent[63–65], and genes associated with cell cycle delay were upregulated at 5 h (Fig. 4B).

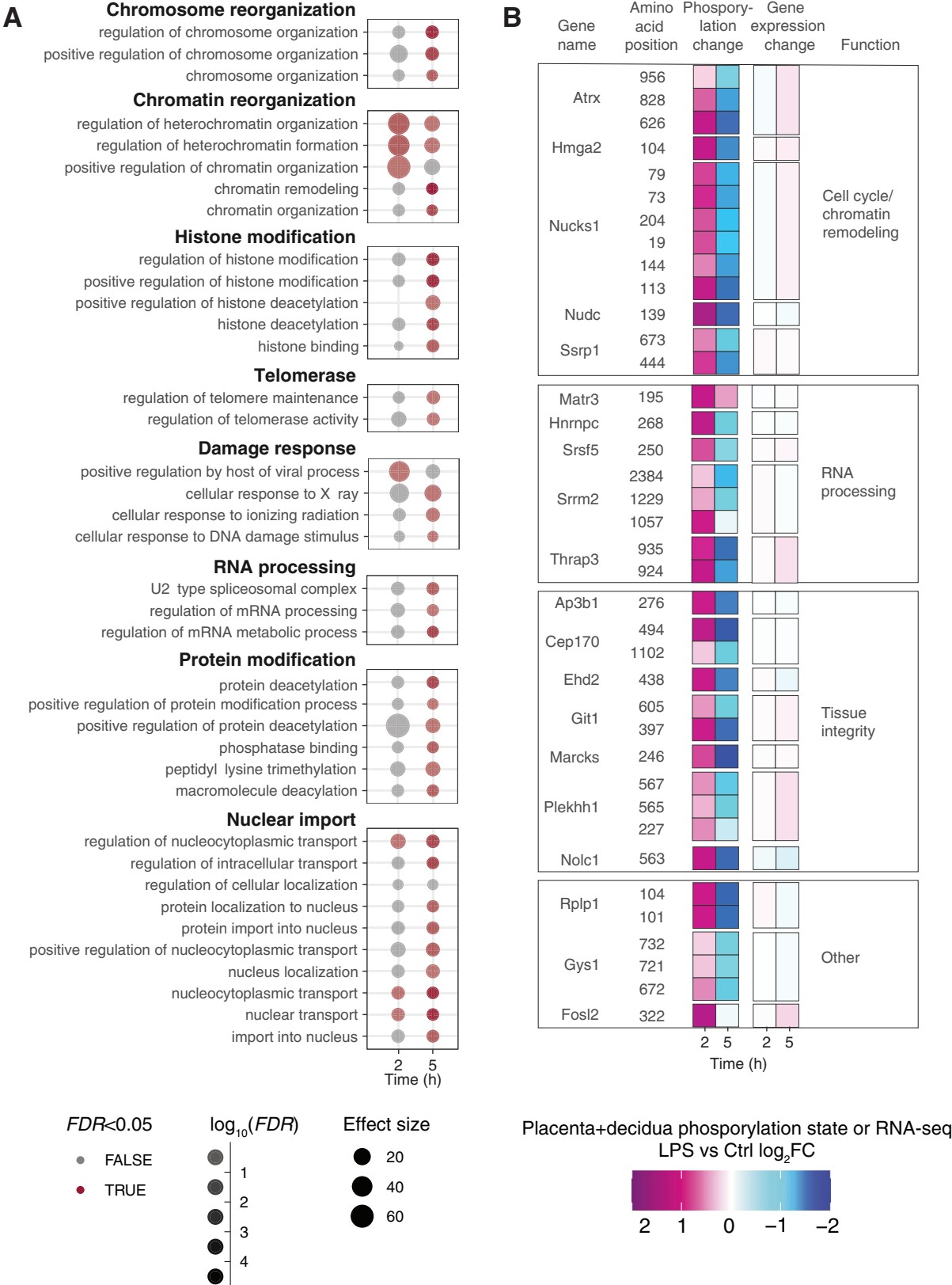

In summary, we observed an early (2 h) increase in levels of phosphorylated proteins with functions similar to those of the differentially expressed genes DE at 5 h. Thus, the placenta+decidua responds with phosphorylative action already at 2 h, before regulation of gene expression at 5 h, indicative of an acute response not immediately visible at the mRNA expression level.

**Pulmonary LPS instillation alters fetal-liver metabolism**

In the fetal liver, 755 genes were DE ($FDR < 0.05$, |$log_2FC$| >0.5) at ≥1 timepoints. We observed several similarities to the placental response. First, most genes were DE only at one timepoint, and 5 h had the largest number of DE genes (473, Supplementary Fig. 1G). Second, GO analysis showed distinctive functional enrichments for each timepoint. As

**Fig. 6 | Phosphoproteomics analysis of early placental response. A** Gene ontology (GO) analysis based on genes with changing phosphosites in placenta. Rows show GO terms, organized in functional themes as indicated on top of boxes. Columns indicate time after instillation (h). Dot opacity indicates significance (red color indicates *FDR* < 0.05), dot size indicates effect size. The analysis is based on proteins having one or more significant phosphosite changes, regardless of the direction of change. **B** Proteins and phosphosites with lipopolysaccharide (LPS)-induced phosphorylation increase at 2 h and decrease at 5 h. The left two heatmap columns show phosphosites for a given protein at 2 and 5 h, as indicated by the two first columns. Colors show average LPS vs. Ctrl log$_2$FC, where positive values represent a gain of phosphorylation in LPS vs. Ctrl. The two heatmap columns to the right show a change in expression level LPS vs. Ctrl log$_2$FC for the corresponding gene at 2 and 5 h. The last column shows the gene function category, based on gene ontology analysis. Displayed phosphosites have a significant (*FDR* < 0.05, see "Methods" section) change in phosphorylation state at 2 or 5 h, an LPS vs. Ctrl log$_2$FC > 0 at 2 h, and higher LPS vs. Ctrl log$_2$FC at 2 than at 5 h. Source data in folder fig6_phospho_placenta and fig1_limma_results_no_maternal_contrasts.csv.

above, we grouped related GO terms into GO themes (Fig. 7A–E, left), and visualized expression changes of top DE GO-linked genes (Fig. 7A–E, middle). Notably, we observed no enrichment of immune response terms.

One large pattern was reminiscent of the placental response: early downregulation of 'energy production' and 'cell cycle regulation', followed by upregulation of 'energy production' and 'biosynthesis' themes at 24 h. Specifically, the fetal-liver upregulated genes associated with negative regulation of cell cycle at 2 h (Fig. 5A, left), including genes with roles in DNA repair (e.g. *Fancm, Neil3, Blm, Rad51ap1, Paxip1*) and genome or chromosome stability (e.g. *Brip1, Fancd2, Hmgb1, Nbn, Fam111a*). Some of these genes were also upregulated at 5 h (Fig. 7A, middle). Genes annotated with GO terms related to 'energy production', including cellular and aerobic respiration, oxidative phosphorylation, and mitochondrial processes (Fig. 7B, left) were subtly downregulated at 2 h, at similar expression levels as Ctrl at 5–12 h, and substantially upregulated at 24 h (Fig. 7B, middle). These genes included *Nduf-*, *Cox-*, and *Atp-*family genes involved in the respiratory chain. Related to this, at 24 h, biosynthesis (including translation, RNA processing, ribosome biogenesis/assembly) and protein/amide synthesis GO terms were upregulated, e.g. large and small ribosomal protein subunit genes (*Rpl-* and *Rps-*genes) and cell cycle progression genes (e.g. *Ube2s, Ctdp1*; Fig. 5C, middle).

Overall, these observations imply that the fetal liver responds to LPS by depleting/inhibiting the translational machinery, cell division, and energy production capacity at 2 h, followed by a gradual increase of these processes until at 24 h where the cell cycle machinery is restored, and energy and RNA/protein/ribosome biosynthesis gene expression is highly increased (Fig. 7A–C, right). We hypothesized that at 24 h, the fetal liver reestablished high capacity for biosynthesis and energy production, to compensate for the decreases in cell cycle progression and biosynthesis at earlier timepoints. This pattern resembles that of the placenta, although placental upregulated genes covered a broad range of functions related to protein synthesis and ribosome biogenesis, while fetal-liver upregulation related almost exclusively to ribosome biogenesis and mitochondrial energy production.

One gene expression pattern unique for the fetal liver was the upregulation of 'catabolic processes' at 5 h and the downregulation of 'metabolic processes' at 12 h. Specifically, genes associated with amino acid breakdown (e.g. *Tat, Sds, Got1, Dao*, and *Asns*,) lipid conversion (e.g. *Lpin1-2*), fatty acid oxidation (e.g. *Sirt2, Eci2*, and *Acox1*), and lipolysis (e.g. *Faah, Pnpla2*, and *Acot8*) were upregulated at 5 h. Many of these genes remained slightly upregulated at 12 h (Fig. 7D, middle). At 12 h, genes associated with metabolism-related GO terms were downregulated, including genes with key roles in steroid/fatty acid metabolism (e.g. *Fads2, Hmgcs1, Hmgcr, Acly, Insig1, Acat2*), lipid synthesis (e.g. *Slc27a3, Gpat4, Ppard, Acsl3, Fads1*) and regulation of fatty acid elongation (e.g. *Hacd3, Hsd17b12*, and *Elovl* family genes, Fig. 7E, middle). Many of these genes use acetyl-/Acyl-CoA for the synthesis of lipids. Acetyl-CoA is synthesized in mitochondria during cellular respiration and cellular respiration genes were downregulated at 2 h (Fig. 7B). Acetyl-CoA availability is determined by the metabolic status of the cell: during fasting, more Acetyl-CoA is needed for ATP generation in mitochondria and less is available for cytosolic lipid synthesis[66].

We hypothesized that the observed pattern reflected a substantial change in fetal metabolic state. The fetus does not synthesize glucose but depends on the maternal supply[67,68]. Similarly, maternally derived EFAs and long-chain polyunsaturated fatty acids (LC-PUFA), in particular DHA, are essential for fetal brain development[67,68]. We suggest that the fetus' availability of glucose and LC-PUFA was briefly altered as an indirect effect of maternal inflammation and net weight loss in LPS-instilled dams vs. Ctrl (Supplementary Fig. 1D). Fetal energy metabolism is remarkably flexible and can switch from glucose to e.g. uptake of lipids, during maternal nutrient deprivation[23]. The mothers' inflammation may cause a shift in the fetal liver to alternative pathways to glucose oxidation for energy generation. Possibly related to these metabolic adaptations, genes associated with glycosylation and development GO terms were downregulated at 5 h (Supplementary Note 3–4, Supplementary Fig. 6A, B).

## Analysis of fetal-liver lipid metabolism after LPS exposure

We reasoned that changes in expression of genes related to lipid metabolism in the fetal liver may be reflected in levels of lipid abundance. Liquid chromatography–mass spectrometry (LC/MS) lipidomics analysis of fetal-liver samples from all timepoints detected 1046 lipid species, where 102 differed significantly between LPS and Ctrl (*FDR* < 0.05, ebayes test[69], Benjamini–Hochberg correction**) at ≥1 timepoints. A heatmap of the 50 most significantly changing lipids revealed a strong pattern (Fig. 8A) where levels of a large set of lipids containing the 18:2 (e.g. linoleic acid (LA) 18:2n-6), 18:3 (e.g. alpha-linolenic acid (ALA) 18:3n-3) and, most prominently, 22:6 (e.g. DHA, 22:6n-3) chains were increased in LPS vs. Ctrl at 12 and, to some degree, 24 h. As we cannot precisely determine the location of the lipid double bonds, these chains may include other FA chains than LA, ALA, and DHA mentioned above. Because of this, we will use '22:6' when referring to the lipid species detected in our data, and 'DHA' when we refer specifically to 22:6n-3. Importantly, lipid species in the LC-PUFA synthesis pathway such as arachidonic acid (ARA, 20:4n-6) and DHA, also known as essential fatty acids (EFA), cannot be synthesized de novo by mammals and must therefore be obtained from the maternal diet, or via essential precursors (e.g. LA and ALA) in the LC-PUFA pathway[70].

Lipids containing 22:6 were primarily triglyceride (TG)–species with 1–3 22:6 chains. Levels of other lipids with one or more 22:6 side chains were also increased, including diglycerides (DG), phosphatidylglycerol (PG), phosphatidylcholine (PC), phosphatidylethanolamines (PE) and fatty acyl esters of hydroxy fatty acid (FAHFA) (Fig. 8A). None of the lipids that were downregulated at all timepoints contained 22:6. More generally, when analyzing all detected lipids, we found that increase in 22:6-containing lipids in LPS vs. Ctrl was most substantial at 12 h ($P = 1.281e-08$, two-sided Whitney–Mann test, Fig. 8B). Our observations suggest that at 12–24 h, peaking at 12 h, the fetal liver contains TG and phospholipids with an increased proportion of 22:6 chains in response to LPS, and also increases the abundance of 22:6 precursors.

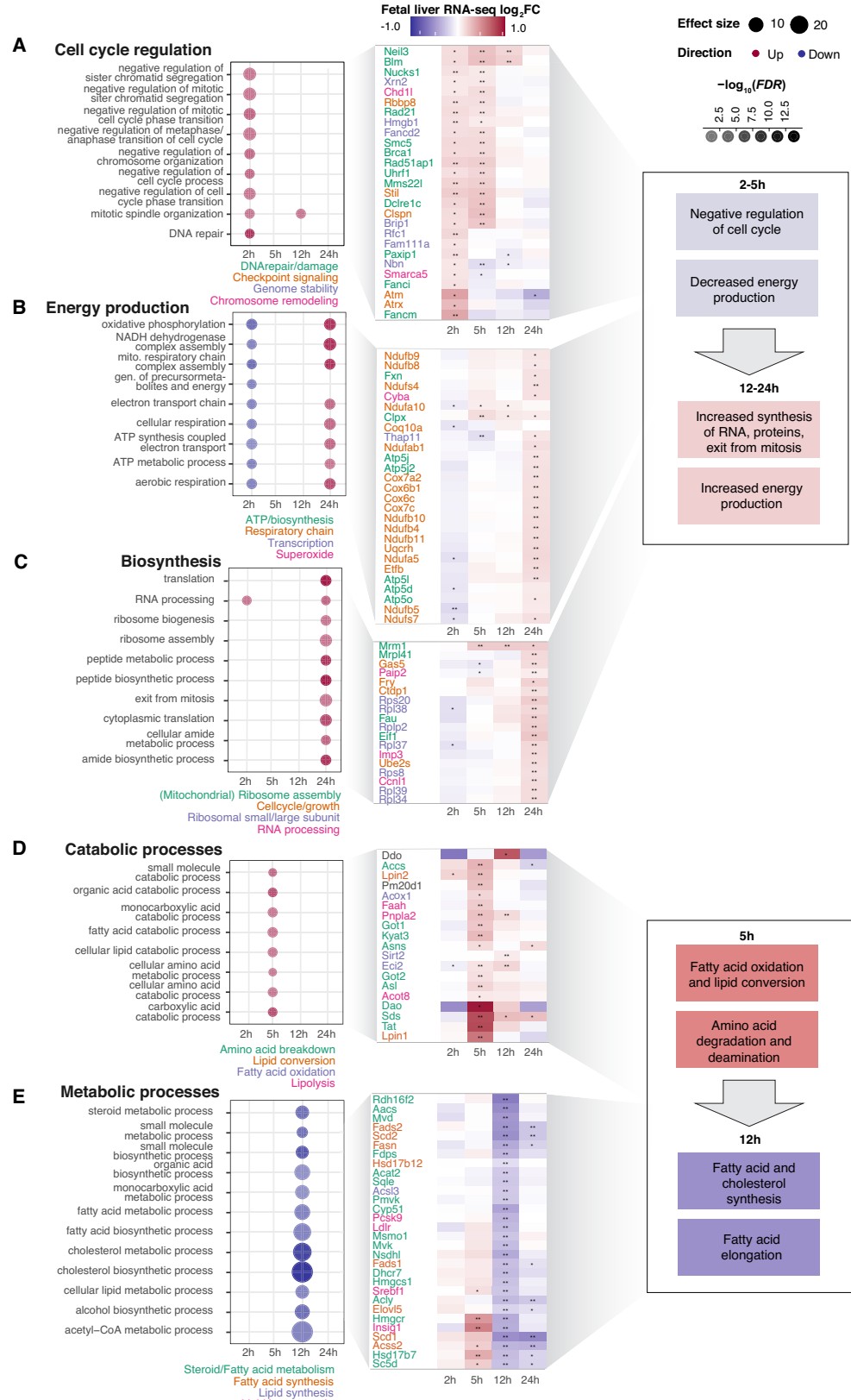

**Fig. 7 | Detailed analysis of fetal-liver gene expression change. A–E** GO term over-representation and expression of linked genes for different GO themes. The figure is organized as Fig. 4A–F, but based on fetal-liver RNA-seq data. Boxes to the right summarizes expression changes early and late in the time course. Source data in fig1_limma_results_no_maternal_contrasts.csv and GO-analysis data in folder GO_enrichment_results_RNAseq.

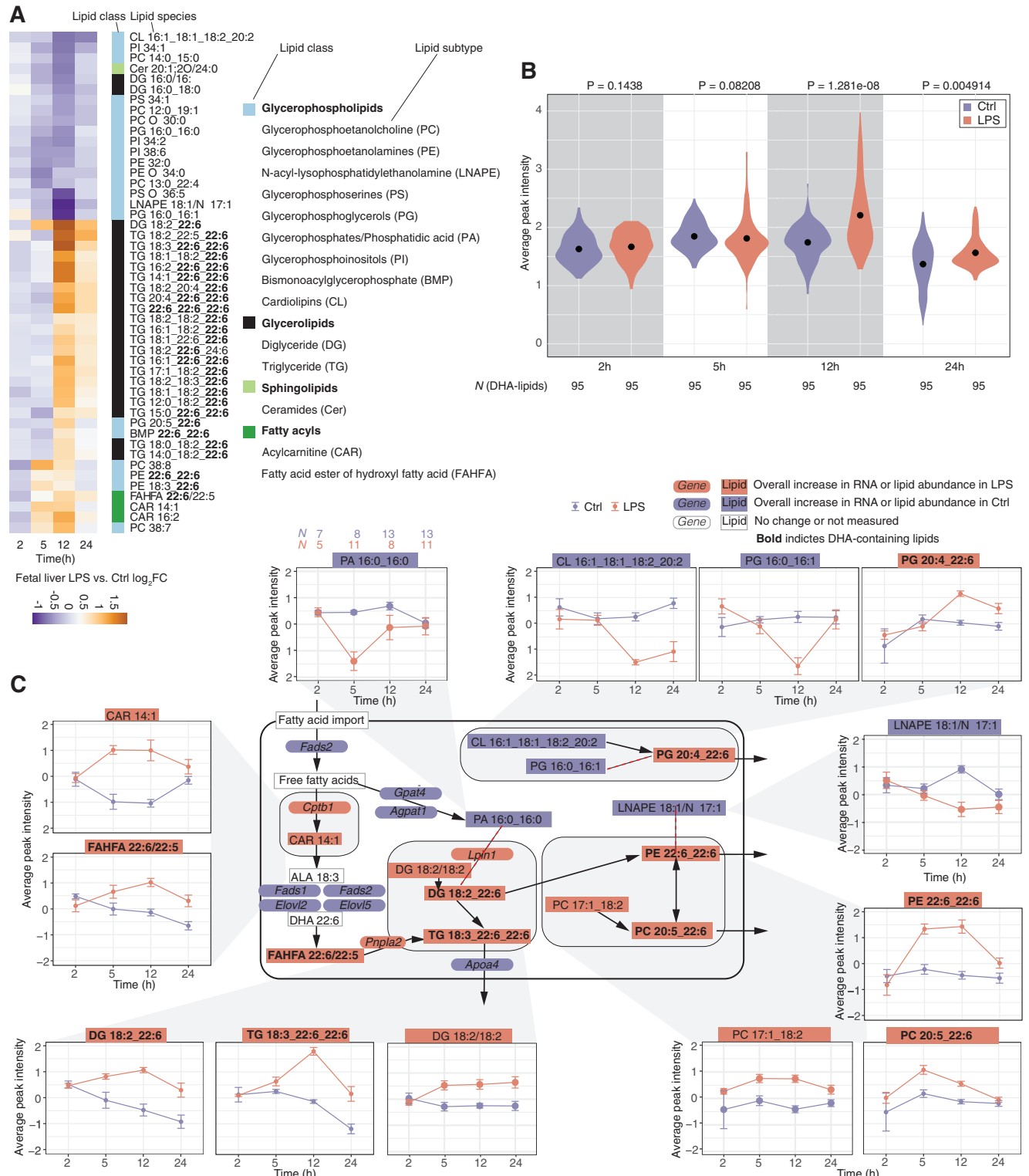

To complement the above with a bottom-up integrative approach, that also integrates the corresponding RNA-seq data, we used DIABLO discriminant analysis[71] from the mixOmics R package[72] that aims to identify a 'multi-omics signature' (in our case based on lipidomics and transcriptomics) that discriminates LPS vs. Ctrl fetal-liver samples. Supplementary Fig. 7A, B shows the most informative genes and lipids for the LPS-Ctrl classification at 12 h, expressed as model 'loading values' where high positive values indicate that high abundance of a gene or lipid is a signature of Ctrl samples while high negative loading values indicate that high abundance of a gene or

lipid is a signature of LPS samples. The topmost informative lipids all had high negative loading values and were all triglycerides containing 22:6 sidechain(s) (Supplementary Fig. 7A), agreeing with our analysis above based on lipidomics data alone and a differential expression framework (Fig. 8A, B). The topmost informative genes had both positive and negative loading values (Supplementary Fig. 7B) but 14/16 genes were associated with the 'Primary metabolic process' GO term (which includes anabolic and catabolic processes), and 8/16 were associated with the 'Lipid metabolic process' GO term, agreeing with the functional enrichment analysis of differentially

**Fig. 8 | Analysis of fetal-liver lipid metabolism. A** LPS vs. Ctrl lipid abundance change at each timepoint. Heatmap is based on the top 50 most significantly lipopolysaccharide (LPS) vs. Ctrl changing lipid species (sorted by lowest *FDR* at any timepoint) clustered by Euclidean distance. Lipids are annotated by class (right). Bold typeface indicates 22:6 side chains. Heatmap cell colors indicate LPS vs. Ctrl log$_2$ fold change (log$_2$FC). Columns right of the heatmap show lipid species class as colored rectangles and name. For class color schema and abbreviations, see the legend to the right. **B** Abundance of 22:6 lipid species in fetal liver. The Y-axis shows the average peak intensity. X-axis indicates time after instillation (h). Numbers on top are *P*-values from Mann-Whitney two-sided tests, comparing LPS vs. Ctrl. Dots within density plots indicate distribution means. Color indicates treatment. **C** Network and abundance changes of selected lipids and lipid-processing enzymes.

Arrows show conversions of lipids or lipid classes (rectangular boxes: bold typeface indicate 22:6-containing species). Selected enzymes for conversion are also shown (rounded boxes). Boxes are colored by whether the lipid or enzyme is up, down, or unchanged over time following LPS instillation. Examples of temporal regulation on lipid species levels are shown as callouts: in these, Y-axis shows average abundance (peak intensity), X-axis shows time after instillation. Error bars show the standard error of the mean. Arrows ending outside the cell boundary denote putative export of lipids into the bloodstream by lipoproteins. Sample size for each group/time-point shown on top left plot (same in all plots, colored by treatment type as above). Source data in folder fig8_9_lipidomics and fig1_limma_results_no_maternal_contrasts.csv.

expressed genes identified in our analysis of RNA-seq data alone (Fig. 7).

We drew a simplified network of lipids with high abundance changes and associated lipid processing enzymes and plotted lipid abundance (Fig. 8C, callouts for selected lipids) and gene expression (Supplementary Fig. 7C) change over time. This enabled six observations:

First, lipids not containing precursors for 22:6-synthesis were typically downregulated in LPS vs. Ctrl at later timepoints, with a reciprocal increase of downstream 22:6-containing lipids (e.g. PG 16:0_16:1 and PG 20:4_22:6, Fig. 8C top). Second, enzymes for elongation of FA into phosphatidic acid (PA) (PA 16:0_16:0), a key inter-mediate in de novo lipogenesis, including *Gpat4 and Agpat1*, were downregulated in LPS vs. Ctrl (Fig. 8C center, Supplementary Fig. 7C). Third, enzymes involved in TG-synthesis were upregulated in LPS vs. Ctrl (e.g. *Pnpla2, Lpin1*; Fig. 8C, Supplementary Fig. 7C). Fourth, lipids with precursors for 22:6-synthesis were upregulated in LPS vs. Ctrl (e.g. PC 17:1_18:2 and DG 18:2/18:2) (Fig. 8C). Fifth, enzymes of the LC-PUFA pathway (i.e. DHA-biosynthesis) (*Fads1, Fads2, Elovl2*, and *Elovl5*) were downregulated in LPS vs. Ctrl (Fig. 8C, Supplementary Fig. 7C). Sixth, lipids with 22:6 chains (with the majority also containing 18:2 and 18:3), which can be stored in lipid droplets or exported into the bloodstream via lipoproteins, were upregulated at 5–24 h in LPS vs. Ctrl, commonly most pronounced at 5–12 h (Fig. 8C).

This pattern of depletion of lipids with medium and long side chains that were not 18:2 or 18:3 (i.e. EFA), followed by an increase in 22:6-containing lipids at 5–24 h, together with corresponding changes in expression of genes associated with TG-synthesis, suggests that one source for the increase in 22:6-containing lipids after LPS instillation was locally stored lipids with 18:2 or 18:3 chains that were converted into 22:6 chains. Supporting this notion, fetal-liver biosynthesis of DHA becomes increasingly important in late pregnancy, to compensate for the increasing discrepancy between maternal supply and increased fetal demand[70], and to buffer diurnal fluctuations of maternal supply[23,70,73]. Maternal LPS exposure may activate both processes. The decrease in LC-PUFA pathway enzymes at 12 h (Fig. 8C, Supplementary Fig. 7C), makes it possible that the increased proportion of 22:6-containing lipids occurs via the conversion of existing lipids with EFA, e.g. DG, TG, and PE. Lastly, fetal-liver lipid stores may deplete and the increase in 18:2-, 18:3- and 22:6 chains may originate from maternal-fetal transfer. This is discussed in the next section.

### Analysis of maternal and fetal lipid metabolism crosstalk

As discussed, the enrichment of 22:6-containing lipids in the fetal liver following LPS instillation could originate from the conversion of stored lipids with EFA within the fetal liver. Alternatively, increased 22:6 levels may originate from increased transfer of EFAs or 22:6 chains from maternal organs such as the liver.

To investigate the latter hypothesis, we examined whether LPS instillation induced lipid abundance changes in maternal liver and/or maternal plasma (fetal plasma measurements were not possible due to

small amounts of fetal blood) by LC/MS lipidomics analysis at all timepoints.

In the maternal liver, we detected 1494 lipid species, where 391 differed significantly between LPS and Ctrl (*FDR* < 0.05, ebayes test[69]) at ≥1 timepoints. A heatmap of the 50 most significantly changing lipids across time (Fig. 9A) revealed a response pattern reminiscent of the fetal liver (Fig. 8A), dominated by LPS-upregulated 22:6-containing lipid species at 12 h, and downregulation of species without 22:6. The same analysis of maternal plasma revealed 1317 lipid species, where 308 differed significantly between LPS and Ctrl at ≥1 timepoint. Plasma data showed a slightly different pattern than the liver: the largest lipid abundance changes also occurred at 12 h, with comparable numbers of up- and downregulated lipid species, where both groups contained, but were not dominated by 22:6-containing species (Fig. 9B). When all detected 22:6-containing species were assessed at each timepoint, maternal liver from LPS-treated mice had higher abundance at 5, 12, and 24 h (*P* < 5.85e-6, two-sided Mann-Whitney tests) compared to Ctrl mice. The change was most pronounced and most significant at 12 h (*P* < 2.2e-16, two-sided Mann-Whitney test), agreeing with the analysis in fetal liver (Fig. 7A); curiously, this was partly due to an overall lower level of DHA-containing species in Ctrl mice at 12 h compared to other timepoints (Fig. 9C). This may reflect that under homeostatic condi-tions, the abundance of certain side chains (e.g. 22:6) is controlled by fasting-feeding cycles and diurnal remodeling of the liver lipidome[74], and that the mice from the 12 h group were killed 2 h into the dark period, that is after 12 h in the light phase with low food intake. The 22:6 abundance increase in plasma of LPS mice was only significantly higher at 12 h (*P* = 0.00063, two-sided Mann-Whitney test, Fig. 9D). Thus, like in fetal liver, LPS instillation increased the abundance of DHA-containing species, with the largest changes observed at 5–12 h in maternal liver, a pattern that was also reflected in maternal plasma, albeit less dominantly.

The lipid abundance change observations were also supported by maternal liver mRNA expression changes at 12 and 24 h, where GO terms associated with metabolic processes such as lipid modification and catabolism of fatty acids were enriched for upregulated genes, while biosynthesis of long-chain fatty acids was dominated by down-regulated genes (Supplementary Fig. 2B). However, the overall maternal liver gene expression response was characterized by upre-gulation of innate immune response genes (Supplementary Fig. 8A and B shows a DIABLO-based integrative analysis of lipidomics and RNA-seq data at 12 h that identified abundance of DHA-containing species as predictive for LPS/Ctrl classification together with, curiously, ECM-associated genes).

EFA and DHA in maternal liver and bloodstream exist as esterified lipids, such as di/triglycerides and phospholipids contained in lipo-proteins. To transfer to the fetal bloodstream[75,76] the chains are hydrolyzed into free fatty acids (FFA) and taken up by the placenta. Maternal-fetal transfer of FAs is estimated to take up to 12 h in humans[75–77], but may be faster in mice. Thus, if the LPS-induced increase of 22:6-containing lipids in the fetal liver at 12 h originates

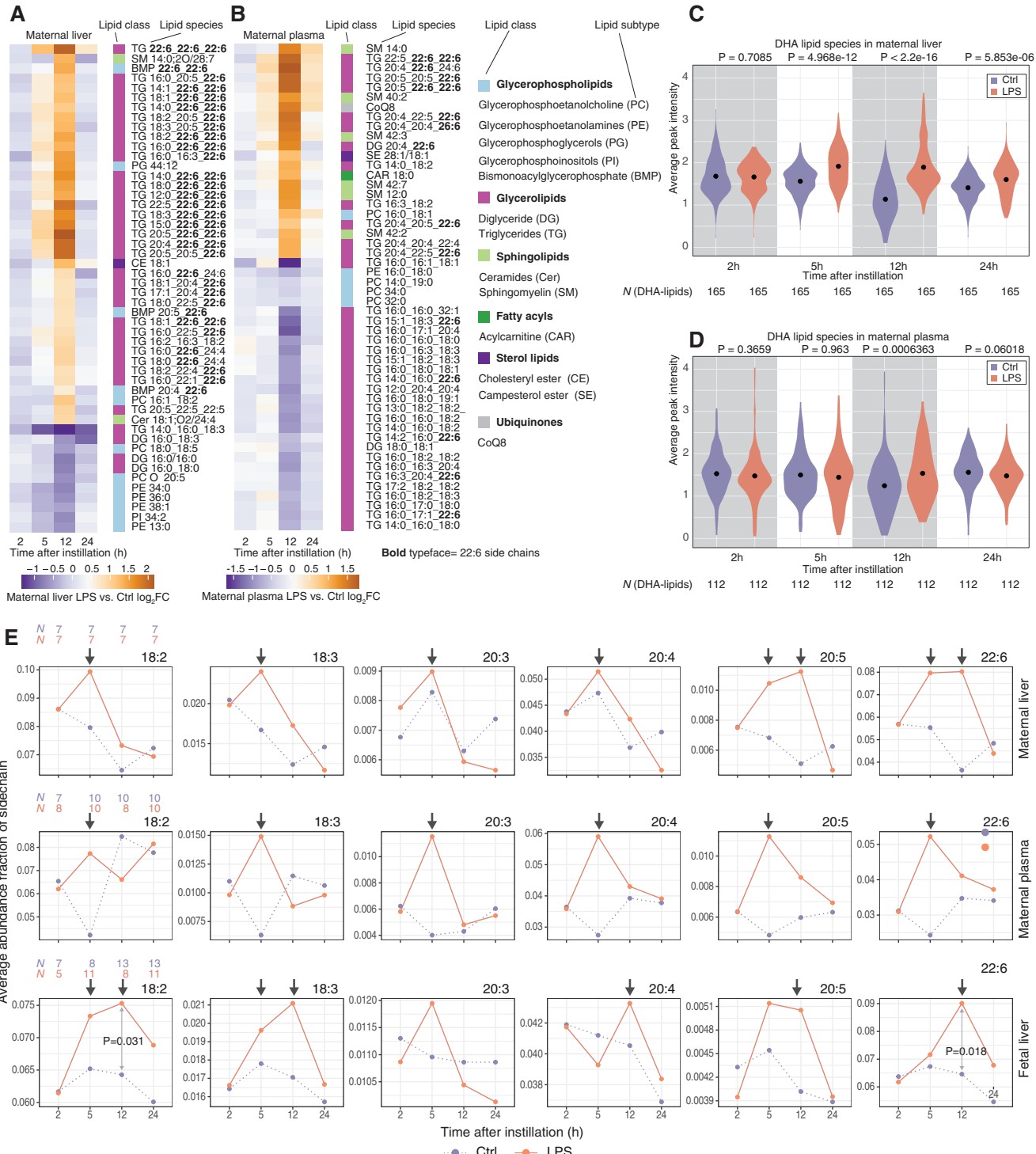

**Fig. 9 | Comparisons of lipid abundance between maternal liver, maternal plasma, and fetal liver. A** LPS vs. Ctrl lipid abundance change at each timepoint for maternal liver. Heatmap is organized as Fig. 8A, showing the top 50 most significantly lipopolysaccharide (LPS) vs. Ctrl changing lipid species in the maternal liver. **B** LPS vs. Ctrl log₂FC of lipid abundance at each timepoint for maternal plasma. Heatmap is organized as Fig. 8A, showing the top 50 most significantly LPS vs. Ctrl changing lipid species in maternal plasma. **C** Abundance of DHA-containing lipid species in maternal liver. The plot is organized as in Fig. 8B but shows maternal liver data. Numbers on top are *P*-values from Mann-Whitney two-sided tests, comparing LPS vs. Ctrl. **D** Abundance of DHA-containing lipid species in maternal plasma. The plot is organized as in Fig. 8B but shows maternal plasma data.

Numbers on top are *P*-values from Mann-Whitney two-sided tests, comparing LPS vs. Ctrl. **E** Abundance change of selected lipid chains over time in maternal liver, maternal plasma, and fetal liver. The Y-axis shows the estimated relative abundance of each lipid chain from lipidomics data, averaged over replicates (see "Methods" section). The X-axis shows time after instillation (h). Color indicates LPS or Ctrl mice/fetus. Side chains are indicated on top of subplots. Each row of subplots shows data from one tissue. *P*-values from Whitney–Mann two-sided tests between LPS and Ctrl at a given timepoint (gray arrow) are shown if *P* < 0.05. Black arrows highlight abundance changes that are specifically discussed in the main text. Sample size for each group/timepoint shown on the leftmost plot in each row, colored by treatment type as above. Source data in folder fig8_9_lipidomics.

from the mother, via transfer of EFA/DHA, there should be an increase in these lipid chains at the latest at the 5 h timepoint in the maternal organs for them to reach the fetal liver at 12 h.

Because the placenta transfers lipids as FFAs, it is relevant to compare the abundance of distinct lipid chains contained in the detected lipid species between tissues. For each lipid species detected in a sample, we computationally extracted its FA chains and estimated their abundance fraction, based on the average abundance of the host lipid(s) across replicates (see "Methods" section). Since only EFA and intermediates in the LC-PUFA pathway serve as basis for synthesis of DHA, we plotted the average abundance fraction of 22:6 and (gamma) linoleic acid (18:2, 18:3), dihomo-gamma-linoleic acid (20:3), arachidonic acid (20:4) and eicosapentaenoic acid (20:5) over time in LPS vs. Ctrl in maternal liver and plasma and fetal liver (Fig. 9E). This led to three observations:

First, in maternal liver from LPS-treated mice the abundance fraction of all chains increased sharply already at 5 h followed by a sharp decrease at 12 h to levels similar or slightly higher than in Ctrl for all chains, except 20:5 and 22:6 whose abundance fractions increased even more at 12 h. In the maternal liver, the proportion of all chains incorporated into lipids such as TG and phospholipids (Fig. 9A) increased at 5 h, while a parallel increase in the conversion of 18:2 and 18:3 to 20:4 and 22:6 also occurred, leading to a peak in fraction abundance of 20:5 and 22:6 at 12 h (Fig. 9E). This would explain the decrease of the 18:2 and 18:3 fractions at 12 h as they were substrates for the conversion. Second, in maternal plasma, the fractions of all chains increased at 5 h but decreased at 12 h after LPS instillation, compared to Ctrl. The decrease could indicate the clearing of these chains from the plasma to increase their bioavailability for maternal and fetal organs. Third, in the fetal liver, all chains (except 20:3) were highly increased at 12 h following LPS treatment. This could indicate synthesis of 20:4 and 22:6 from 18:2 and 18:3 already present in the fetal liver, and a resulting increase in the proportion of 22:6 chains.

In summary, in maternal liver and plasma, the largest abundance change for many chains following LPS treatment occurred prior to 12 h, most often at 5 h. In fetal liver, the largest change was observed at 12 h for both DHA and precursors. In other words, there was a temporal shift in abundance between maternal and fetal tissues that is compatible with the time needed for materno-fetal transfer. Hence, it is likely that both the maternal and fetal liver respond to maternal lung LPS exposure, by increasing the abundance of lipids with EFA and DHA, such as TG and PE, which become available for other maternal/fetal organs via secretion into the bloodstream, where a similar increase was observed in the maternal plasma. The need to increase EFA and DHA bioavailability may be due to limited food intake in LPS mothers, as this would reduce maternal uptake of EFA and DHA from diet, and subsequently reduce their supply to the fetus. Overall, the results from Figs. 8 and 9 suggest that both the maternal and fetal metabolic alterations increase bioavailability of DHA and its precursors, and the data supports both of the non-mutually exclusive hypotheses raised above: (i) conversion of already existing EFA into DHA/22:6-containing lipids in fetal liver, and (ii) transfer of EFA/18:2, 18:3 and DHA/22:6 through the placenta following increased bioavailability in the mother.

## Discussion

Here, we analyzed responses to acute pulmonary maternal inflammation, across maternal, placental, and fetal tissues and time, based on transcriptomics, phosphoproteomics, and lipidomics, complemented by targeted protein abundance assays of maternal plasma and imaging. Our main findings and interpretations are summarized in Fig. 10.

While LPS induced a strong innate immune response in the maternal lung and liver, through activation of TLR4-pathway genes and downstream signaling pathways (Fig. 1B, C), and increased levels of cyto- and chemokines in maternal plasma (Fig. 2A), this did not extend to placenta and fetal liver; these displayed functionally distinct and temporally dynamic responses. Specifically, the placenta increased expression of tissue-integrity genes while DNA/RNA processing, biosynthesis, and energy production gene expression decreased at 5–12 h, followed by compensatory increases in protein synthesis and ribosome biogenesis gene expression and, at 24 h, upregulation of genes associated with mild ER stress. The placenta did not mount an innate immune response, even though it has the ability to do so in contact with LPS[36].

SOCS3, crucial for placental development[78], and a key inhibitor of the IL-6-Jak/STAT-pathway, was upregulated at both mRNA and protein level at 5 h in the placenta and less in decidua following LPS instillation and could act to inhibit the induction of inflammatory cascades. There was an increase in SOCS3 protein expression, which was particularly intense in the spongiotrophoblast of the junctional zone that serves as a separator of maternal decidua and the labyrinth zone, and as structural support of the latter. Overexpression of SOCS3 in the junctional zone could thus partially protect the labyrinth from activation of the IL-6-Jak/STAT-pathway via IL-6. However, initiation of labor at GD20 of mouse pregnancy requires activation of pro-inflammatory cytokines[79], whereas SOCS3 overexpression tunes the immune system towards an anti-inflammatory profile. Reduced SOCS3 expression in the placenta at parturition has been suggested to reduce its inhibition of the switch towards pro-inflammatory processes[80], while SOCS3 is overexpressed in infection-related preterm labor, possibly to attenuate its adverse effects[47]. The overexpression of placental SOCS3 in response to LPS could therefore serve to limit initiation of pro-inflammatory cytokine signaling and consequences such as preterm labor, but also affect activation of term labor, both with adverse effects for mother and fetus. Overall, the higher expression of Il6ra and Il6st in the decidua, and the placenta-specific increase in Socs3/SOCS3 expression, demonstrate a capacity of selective 'immune inertia', and may be a feature of placental immune adaptation[81] and tolerance[82,83].

Reminiscent of the placenta, the fetal liver did not activate an immune response and instead first downregulated cell cycle signaling and growth genes and then increased growth and protein synthesis gene expression at 24 h. A strong pattern was the early and pronounced differential expression of metabolism- and catabolism genes, especially increases in enzymes catalyzing lipid conversions and lipolysis and decreases in enzymes synthesizing fatty acids and de novo lipogenesis. Lipidomics analysis revealed that at 12 h, the proportion of 18:2-, 18:3- (i.e. EFA), and especially 22:6 (i.e. DHA)-containing TGs and PEs were increased, while lipids without these chains were decreased. In the fetal liver, deposits of DHA-containing lipids and elongation or desaturation of existing lipids are suggested to buffer decreased availability from the mother[73] during late pregnancy. It is therefore likely that there is an increased internal mobilization and enrichment of fetal-liver DHA pools into carriers such as TG.

Maternal liver and plasma also increased proportions of EFA- and DHA-containing lipids, but slightly earlier than fetal liver, peaking at 5 h. Maternal-fetal transfer of FFAs can take up to 12 h, and this makes it possible that the increased proportions of DHA-containing lipids in fetal liver at 12–24 h were partially supplied from the mother. These processes enhance the bioavailability of DHA for both mother and fetus. A determination of the degree of maternal-fetal transfer of EFA/DHA and their accretion by fetal tissues would require the administration of labeled EFA/DHA and the subsequent tracing of their abundance and incorporation into lipids in fetal organs[76].

During homeostasis, the maternal liver metabolism is regulated diurnally and alternates between periods of feeding and fasting[74]. Adaptation to fasting involves the release of lipolysed TGs from adipose tissue to the circulation, and uptake by the liver where FAs are utilized for energy production and synthesis of TGs that are incorporated into lipoproteins and reenter the circulation, for supply to tissues, such as the brain. Importantly, during fasting, there is no de novo

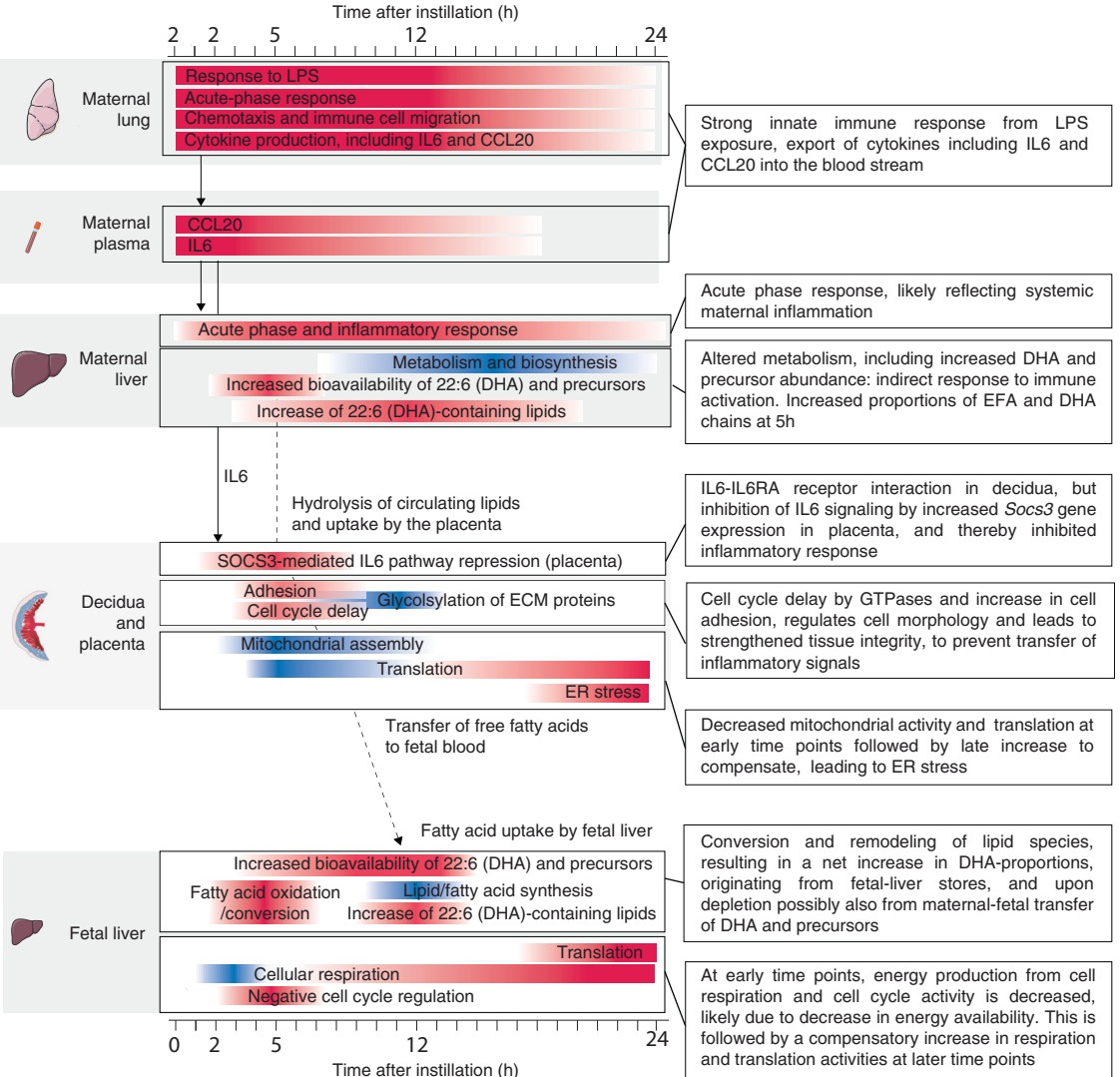

**Fig. 10 | Overview of main findings and interpretations.** The X-axis shows time after installation (h). Larger gray areas indicate changes in respective tissue, as denoted by cartons to the left. Bars within gray areas denote classes of genes or lipids. Red color indicates upregulation following lipopolysaccharide (LPS) treatment, blue indicates downregulation. Arrows between gray areas denote potential tissue interactions through signaling molecules (e.g. cytokines) or the transfer of molecules (e.g. lipids or fatty acids). EFA essential fatty acids, DHA docosahexaenoic acid, ER endoplasmic reticulum. Artwork adapted from bioicons (https:// bioicons.com/, CC 0 license).

lipogenesis. The liver adaptation to a period of intermittent fasting is characterized by an increased abundance of 22:6-containing lipids and resembles the lipid and chain profile we observed in LPS-exposed mice, i.e. maternal liver and plasma at 5–12 h and fetal liver at 12–24 h (Figs. 8 and 9). The reason why brief maternal inflammation, fever, and related hypophagia induce a metabolic fasting response can be that maternal energy metabolism during late pregnancy can switch rapidly from glucose to fat – termed accelerated starvation – to satisfy the growing fetus' metabolic demands[84], in particular demand of DHA. The fetus accelerates its ability to take up and accumulate maternal lipids in response to fasting[23]. The metabolic adaptations we observe in our study may be within the physiological range, but they might also deplete both fetal and maternal lipid storages, at a time when parturition at GD20 is very near. The first few hours after birth are marked by a surge in adipose tissue mobilization and resulting neonatal brain-accretion of DHA[73]. If fetuses from the LPS-exposed mice do not manage to get beyond the fasting response, the resulting insufficient fetal-liver DHA deposits may decrease DHA accretion by the neonatal brain, potentially causing lasting effects on neonatal brain development[85].

Interestingly, DHA exhibits anti-inflammatory properties and promotes immune functions while inhibiting the production of pro-inflammatory cytokines. A few studies administering high intraperitoneal doses of LPS (50 µg/kg at GD 15–17 and 0.12 µg/g mouse at GD17, respectively) showed decreased levels of DHA in the fetal liver after 24 h and changes in lipid metabolism in the adult offspring[86] and exacerbation of maternal and fetal inflammatory responses in mothers with dietary omega-3 (including DHA) deficiency[87]. In contrast to these results, we observed increased DHA proportions in carrier lipids such as TG and phospholipids in the first maternal liver and plasma (from 5 h) and later fetal liver (12–24 h). This could potentially be due to the lower and pulmonary LPS dose utilized in our study. Nevertheless, increased circulating DHA levels may promote an anti-inflammatory environment, which could partly explain the dampened immune response in the placenta, and thus the lack of transfer of inflammation to the fetus.

Our study has important limitations. First, the timeframe of the outcome assessments was 0–24 h after exposure, and only effects of a single lung instillation with a specific molecule, LPS of E. Coli serotype 00:55 B5, at a certain dose. The LPS dose was chosen to model robust

airway inflammation and at the same time avoid severe outcomes, such as preterm birth. Our data clearly shows that this dose induced a pronounced inflammatory response in maternal lung and liver, and measurable effects in all tissues, including those of the fetus. These effects were recapitulated in two or more (omics) assays. While our dose is physiologically relevant, it is probable that some effects would change with dose, e.g. at substantially higher LPS doses the observed dampening of the placental immune response could be replaced with a measurable inflammatory response. Lung administration induces inflammation at the port of entry and decreases systemic dose rate and absorption due to the passage of the lung barrier. Hence, activation and export of secondary messengers such as cytokines induced by LPS may be more relevant to assess than systemic levels of LPS. Intravenous administration renders LPS immediately available to all organs, via blood, prior to metabolic processing by the maternal liver[39], while intraperitoneal administration renders LPS available for first-pass metabolism in the maternal liver, and hence induce more pronounced inflammation in the maternal liver. Our findings might therefore not translate directly to studies applying higher dose levels and/or other routes of administration. Similarly, many MIA studies used other inflammatory agents such as vira or poly I:C, which display both unique and shared effects with LPS. Lastly, we focused exclusively on responses in female fetuses. MIA is shown to translate into sexually dimorphic responses in the placenta and offspring. Therefore, extrapolation to males should be supported by experimental evidence[88].

As a summary, our study describes the maternal and fetal responses over time to an inflammatory maternal insult in the lung. A key finding was that inflammation did not transfer from mother to fetus, but both mother and fetus responded with metabolic adaptations similar to those observed during starvation.

## Methods

### Inclusion and Ethics
All animal procedures followed the guidelines for care and handling of laboratory animals established by the EC Directive 86/609/EEC and Danish regulation (Danish Ministry of Justice, Experimental Animal Inspectorate, permit 2015-15-0201-00569). The local animal welfare committee (The animal welfare committee of the National Research Centre for the Working Environment) approved the specific protocol prior to the study.

### Animals
80 Nulliparous C57BL/6JRj mice (Janvier, Saint Berthevin Cedex, France) 8–12 weeks old from the same barrier room were time-mated and pregnancy confirmed by the presence of vaginal plug the morning after mating (designated gestation day (GD) 0). Dams arrived at the institute at GD 11 or 12 and acclimated for 6–7 days prior to exposure. Mice were pair housed in clear 1290D euro standard polypropylene cages with Aspen bedding (Tapvei, Estonia), enrichment (mouse house 80-ACRE011, Techniplast, Italy; small aspen blocks, Tapvei, Estonia), and nesting material (Enviro Dri, Lillico, Biotechnology, UK), under controlled conditions (temperature 21–22 °C; humidity 55 ± 10%; ventilation 15–20 air changes/hour; 12 h light-dark cycle with lights on at 06.00 a.m.) and access to food (Altromin 1314 for breeding, Brogaaarden, Denmark) and tap water ad libitum. Animals were weighed on the day of arrival and the day prior to exposure to confirm pregnancy.

### Exposure and dissection of maternal/fetal organs
Lipopolysaccharide (LPS; E. Coli serotype 00:55 B5 LPS (Sigma Lot nr. 025M4040V)) was diluted to the final concentration (0.02 µg/µl) in double distilled pyrogen-free water (Chem-Lab, Zedelgem, Belgium). In the morning of GD17, the pregnant mice were randomized into control and LPS treatment groups (denoted Ctrl and LPS, respectively), distributing weights among the groups evenly. Out of 80 mice in total,

74 were pregnant. Animals were placed in a whole-body inhalation chamber with an attached anesthetic vaporizer (Penlon Sigma Delta, Abingdon, UK), delivering 3–4% isoflurane in filtered air, and were intratracheally instilled with 50 µl of vehicle (Ctrl) or 1 µg LPS in 50 µl vehicle, followed by 200 µl of air. Vehicle and LPS were administered through a 0.58 mm polyethylene tube (Ref: 427411, Becton Dickinson, Brøndby, Denmark) attached to a plastic syringe. The procedure has been shown not to affect gestation, offspring viability nor growth[89]. The first instillation was given at 8 a.m. After instillation, animals were returned to their cage, briefly placed on heating pads, and checked upon regularly until euthanization. At 2, 5, 12, and 24 h, dams were terminally anesthetized by subcutaneous injection of 0.2 ml of Zoletil mixture (tiletamin/zolazepam, xylazin og fentanyl) and killed by exsanguination by withdrawal of heart blood into Eppendorf tubes containing 36 ml K$_2$EDTA ($N$ = 7–9 per exposure/timepoint). The uterus was excised and opened. Fetuses were excised from their embryonic sac, their viability confirmed, killed by decapitation, sexed by visual inspection, and their position in the uterus noted. From each litter, the first female fetus encountered in the right uterine horn, counting from the cervix, was selected and saved for analyses. The placenta was dissected into chorion (chorionic plate, labyrinth, and junctional zones) and decidua by blunt/stump dissection under stereomicroscope (Wild Heerbrugg, Switzerland)[90]. From dams, the liver and right lung were dissected. Dissected organs were snap-frozen in liquid nitrogen and kept at −80 °C. Fetal livers were excised later. Maternal blood was centrifuged at 2000 × $g$ at 4 °C for 5 min and plasma was stored in aliquots at −80 °C until analysis. A maximum of one female fetus per dam was used for any one outcome, except for lipidomics, where two female fetuses were used in some groups/timepoints.

### Fetal sex genotyping
Genotyping of pup sex was performed on DNA extracted from the fetal tail, by PCR using primers for *Ddx3Y* (denoting the Y chromosome in males: forward 5′-GGG TCT GTG ATA AGG ACA GTT CA-3′, reverse 5′-CAC GAC CAC CAA TAC CAT CAT AG-3′) and *Rpl13a* (denoting the X chromosome present both in males and females, forward 5′-AGC CTA CCA GAA AGT TTG CTT AC-3′, reverse 5′-GCT TCT TCT TCC GAT AGT GCA TC-3′), purchased from TAG Copenhagen A/S. Following PCR, the reaction was separated on a 1.5% agarose gel. A band for Ddx3Y of 908 bp, connoted XY whereas the lack of this band but the presence of Rpl13a of 129 bp, connoted XX.

### RNA extraction and library construction
Total RNA was isolated from frozen maternal lung and liver, chorion, decidua, and fetal liver. Briefly, 20–50 mg of tissue was homogenized with a T 10 basic ULTRA-TURRAX® blender (IKA, Staufen, Germany) in 700 µl lysis buffer with 7 µl mercapto-ethanol. RNA extraction was carried out utilizing magnetic beads technology, on a chemagic Prepito® (Perkin Elmer, Waltham, Massachusetts), as recommended by the manufacturer. Concentration and purity were measured on a NanoDrop1000 spectrophotometer, with all samples showing an A260/280 ratio between 1.9 and 2.1. RNA integrity was analyzed by 2100 Bioanalyzer (Agilent Technologies) with Agilent RNA 6000 Pico Kit (Agilent Technologies) as recommended by the manufacturer. All samples used for RNA-seq displayed RNA integrity number (RIN) above 7. cDNA library construction and paired-end sequencing was carried out by Novogene (China). To exclude ribosomal RNA, polyA selection was done.

### RNA-seq analysis
RNA-seq analysis from quality control to DE analysis was made with a Snakemake[91] pipeline using Conda (https://conda.io). Sequencing produced a total of 740 libraries (370 paired-end), with a median read depth of 24 million reads (mean 25 million reads, min 19.9 and

max 46.6), all of which passed initial QC controls performed by multiqc[92]. We detected base-pair over-representation in the first 11 bp of each read, as expected due to biases in random primers, and removed them using the seqtk version 1.2 (https://github.com/lh3/seqtk). We used Salmon[93] to map reads to gencode mouse transcriptome version M23[94], which is an annotation of the genome assembly version GRCm38. For mapping, we created an index using $k$-mer size 31 bp and supplying the genome in order to create decoy $k$-mers to account for biases due to unannotated transcribed genomic regions. We mapped libraries to the resulting index using Salmon quant using the following options: gcBias, seqBias, validateMappings, numBootstraps = 10, and minScoreFraction = 0.8. Finally, we annotated the resulting quant files using the R library tximeta[95] and removed all features that lacked annotation. We performed all initial exploratory analyses and plots using TPM-normalized data and used the raw count data for the differential expression analysis. Before fitting statistical models, we normalized the count data from Salmon using TMM and retained only transcripts with >10 reads in at least 70% of the samples of the same condition[69]. We detected differentially expressed genes using generalized linear models in limma after variance stabilization using voom[69]. We converted all $P$-values to $Q$-values using FDR correction using FDRtool[96]. Due to the large differences in expression between most tissues (Supplementary Fig. 1E) we fitted independent models for each tissue, with the exception of placental samples (see below). For maternal lung, maternal liver, and fetal liver we used the model formula $E = O + timepoint + timepoint{:}LPS$ which estimates each gene's average expression for control samples in each timepoint, and then estimates the difference in expression between control and treatment samples from the same timepoint. Since we found that the placenta (encompassing chorionic plate, labyrinth, and junctional zones) and decidua were very similar in their overall expression response (Supplementary Fig. 2C) we fitted a single model for both, with the formula $E = O + timepoint + maternal + timepoint{:}LPS + timepoint{:}LPS{:}maternal$. This model calculates the average gene expression at each timepoint for control samples of both placenta and decidua and then tests for (1) differences in the expression of decidua and placenta within control samples, (2) shared responses to maternal inflammation and (3) differences between the responses to maternal inflammation of both.

## Gene set enrichment analysis

We performed gene set enrichment using leading-edge analysis using gprofiler2[97]. For each tissue, we selected all genes with baseline normalized fold expression >0 and ordered them by $Q$-values, signed according to up or downregulation (decreasing to test for upregulation, increasing to test for downregulation). We used the list of all genes expressed in each tissue above $\log_2$ normalized counts of zero in at least one timepoint as the background set for all enrichment tests in that tissue. We used the gSCS method for $P$-value correction with a threshold of 0.05 and testing only for over-representation of GO terms. In order to summarize the results for figures, we retained only significant Biological Process terms with more than 10 and less than 1000 terms. Due to the vastly different amount of differential expression between tissues, we used different methods to summarize the results. From the maternal lung and liver GO terms, we curated lists of highly significant (FDR < 0.01 at any timepoint) GO terms that were both interesting and representative, which were used for Figure S1B, C. Similarly, for the placenta and fetal-liver GO terms, we manually curated a list of interesting and representative GO terms (FDR < 0.01 and effect size >2, at any timepoint). From these, we extracted the complete list of genes annotated with the respective GO term(s), and only retained those genes that were differentially expressed (FDR < 0.01) at the timepoint where the GO term was significantly enriched, and from these retained the 50 most significant genes sorted by FDR.

## Cytokine analysis in maternal plasma

A total of 31 chemokines were measured in maternal plasma using a magnetic bead-based kit (Bio-Plex Pro Mouse Chemokine 31-Plex). The Luminex xMAP multiplexing technology and the Bio-Plex® 200 platform (Bio-Rad Laboratories, USA) were used for the analysis of the plasma samples. Plasma was diluted 1:4 and the protocol carried out according to the manufacturer's description. The standard curve was run in duplets, and the samples in singlets. After initial analysis of plasma protein concentrations, 10 out of the 31 chemokines were chosen for further analysis.

## Ligand-receptor analysis

We used the CellPhoneDB database[40] annotations to link annotated secreted proteins/peptides with cognate annotated receptors. The linkage allowed for many-to-one and many-to-many matches (e.g. several ligands bound to one receptor, or vice versa). Since the cellphone database is human-based, we translated the human gene names to their orthologous mouse counterparts using Ensembl[98] annotation.

## Immunohistochemistry

For immunohistochemical staining, placental tissue from female pups was fixed in paraformaldehyde (PFA 4%, overnight) before embedment in paraffin. Sections were cut to a thickness of 3.5 μm for SOCS3 staining and 7 μm for ATF4 staining with a Leica HistoCore AUTOCUT Rotary Microtome (Leica Biosystems, Wetzlar, Germany) and Leica RM2245. For both SOCS3 and ATF4, the remaining experimental protocol was carried out on a Bond RXm (Leica Biosystems, Wetzlar, Germany) staining robot. For ATF4, sections were dewaxed with Bond Dewax solution and boiled for target retrieval in BOND Epitope Retrieval Solution 2 for 20 min. For SOCS3, sections were dewaxed with Bond Dewax solution and rinsed with HIER buffer, and incubated at 100 °C, twice. For ATF-4, endogenous peroxidase activity was blocked with 3% $H_2O_2$ for 5 min. For both ATF4 and SOCS3, sections were incubated in 10% donkey serum (Candor Bioscience, Wangen, Germany) for 10 min (ATF4) or 5 min (SOCS3). The primary antibody ATF4 (1:100, Abcam/ab31390, Anti-rabbit) was incubated 1 h at ambient temperature, while SOCS3 (1:800, Invitrogen/PA5-87485) was incubated 15 min at ambient temperature. For ATF4, sections were coated with a biotinylated secondary antibody (dilution 1:500, donkey anti-rabbit, reference: 711-065-152, Jackson ImmunoResearch) for 30 min and incubated with ABC solution (Vectastain PK-6100; Vector Laboratories, Linaris, Dossenheim, Germany) at room temperature for 30 min. Finally, the AEC kit (SK-4205; Vector Laboratories) was applied, and sections were counterstained with hematoxylin for 15 s and embedded in Vectamount AQ Aqueous Mounting Medium (reference: H-5501, Vector Laboratories). For SOCS3, sections were coated with a biotinylated secondary antibody (dilution 1:500, Anti-rabbit Poly-HRP-IgG) for 8 min, and incubated with mixed DAB Refine for 10 min, then counterstained with hematoxylin for 5 min, and dehydrated in HE Gemini. Slides were scanned using a PANORAMIC SCAN (3DHISTECH Ltd., Budapest, Hungary) (ATF4) or a NanoZoomer-XR Digital slide scanner C12000-01 (Hamamatsu Photonics K.K, D-82211 Herrsching am Ammersee, Germany) (SOCS3), and analyzed with the QuPath 0.5.0 software.

## Proteomics sample preparation, TMT labeling, and chromatography

Female placenta samples from 5 h (16 samples in total) were subjected to lysis with 5% SDS in water at room temperature, sonication, and boiling for 10 min at 95 °C Protein concentration was measured, and 200 μg of the sample was processed to reduction, alkylation, and digestion with LysC and Trypsin enzymes by the Protein Aggregation Capture (PAC) method as in ref. 99 using MagReSyn® HILIC microparticles (ReSyn Biosciences Ltd). 100 μg of tryptic peptides from each sample were used for the TMTpro reagent (Thermo Fisher Scientific)

labeling procedure according to the manufacturer's protocol. The TMT-labeled peptides were pooled together, lyophilized, and applied to a phosphopeptide enrichment protocol[100] by immobilized metal-ion affinity chromatography (IMAC) with MagReSyn® Ti-IMAC magnetic microparticles (ReSyn Biosciences Ltd). The eluted from IMAC peptides were subjected to the High pH fractionation as in ref. 101 resulting into 14 HpH fractions that were dried out in a vacuum centrifuge and resuspended in 0.1% trifluoroacetic acid (TFA) for subsequent LC-MS/MS.

## Phosphoproteomics LC-MS/MS, raw data processing and analysis

The resulted samples were infused into the home-made fused silica column (inner diameter of 75 μm) packed with C18 resin (1.9 μm beads, Reprosil, Dr. Maisch) with an EASY-nLC 1000 ultra-high-pressure system (Thermo Fisher Scientific) for reverse phase chromatography. A high-field asymmetric waveform ion mobility spectrometry (FAIMS Pro) device (Thermo Fisher Scientific) was placed between a nanoelectrospray ion source and an Orbitrap Exploris 480 mass spectrometer (Thermo Fisher Scientific). The FAIMS was operated in a standard resolution mode with Cvs. −50 V and −70 V that were applied to all scans of the entire MS run. The Orbitrap Exploris 480 MS was used in positive-ion mode with a capillary temperature of 275 °C acquiring MS data in a data-dependent mode (DDA) based on cycle time with master scans equal to 1.5 s. Method duration was 180 min with a normalized AGC target value 300% at full MS scan. The resolution was set to 120,000 with a scan range of 400–1400 m/z and maximum injection time (IT) 50 ms. The Normalized Collision Energy (NCE) by HCD was 32%. For the MS/MS scan resolution was set to 45,000, maximum IT to 120 ms, isolation window with 0.7 m/z, normalized AGC target was 200%. The dynamic exclusion window was set to 60 s. The resulting 14 raw files were processed to MzXML files using FAIMS MzXML generator (https://github.com/coongroup/FAIMS-MzXML-Generator) to search 28 MzXML files with MaxQuant (version 1.6.7.0) applying TMTPro correction factors for TMT channels quantitation by the software. The search was done against a target/decoy database (Mus Musculus, SwissProt from September 2019 with 17,013 entries) with *FDR* < 0.01 with the following parameters: main search peptides tolerance was 10 ppm, fragment mass tolerance was 20 ppm. An enzyme for protein digestion was specified as trypsin with allowing two missed cleavages. Cysteine's carbamidomethylation was specified as fixed modification and protein N-terminal acetylation, oxidation of methionine, and phosphorylation on STY residues were set as variable modifications. Results from the MaxQuant search "Phospho (STY) Sites" table were used for identified phosphosites quantitation analysis using the edgeR package[102] using TMM-based normalization and differential abundance analysis using *FDR* < 0.05 as a significance threshold. Gene set enrichment analysis of proteins with changing phosphorylation states was made in the same way as RNA-seq data, with the following changes: as input, we selected all proteins with one or more changing phosphosites (*FDR* < 0.05, as defined above), and for each protein, we only retained the lowest *FDR* value if several sites satisfied this criterion. We then ordered these proteins based on *FDR* and used this list as input for gprofiler2. As background, we downloaded a list of all *M musculus* proteins with one or more experimentally validated phosphosites from the EPSD database version 1.0[103], and then intersected this with RNA expression data from the same tissue, only keeping genes/proteins that were detected by RNA-seq and having one or more phosphosites.

## LC/MS lipid profiling, data processing and analysis

Lipids were extracted from maternal liver, plasma, and fetal-liver samples (20 mg/20 μl) using Folch extraction[104] with 8–12 replicates from each experimental group at each timepoint. Prior to tissue lysis, Splash mix (Merck) was added to the extraction solvent, and tissue

samples (except for plasma) were lysed by beat beating in a FastPrep-24 homogenizer. After centrifugation and phase separation, the apolar and polar phases were transferred to separate tubes, and the apolar phase dried under $N_2$. Samples were resuspended in 30 μl methanol/chloroform (1:1) and centrifuged (5 min/16,000 × g/22 °C) before transferring to HPLC vials. A quality control sample was constructed by pooling 3 μl of each sample. Samples (0.5 μl) were injected using a Vanquish Horizon UPLC (Thermo Fisher Scientific) equipped with a Waters ACQUITY Premier CSH (2.1 × 100 mm, 1.7 μM) column operated at 55 °C. The analytes were eluted using a flow rate of 400 μL/min and the following composition of eluent A (Acetonitrile/water (60:40), 10 mM ammonium formate, 0.1% formic acid) and eluent B (Iso-propanol/acetonitrile (90:10), 10 mM ammonium formate, 0.1% formic acid): 40% B from 0 to 0.5 min, 40–43% B from 0.5 to 0.7 min, 43–65% B from 0.7 to 0.8 min, 65–70% B from 0.8 to 2.3 min, 70–99% B from 2.3 to 6 min, 99% B from 6–6.8 min, 99–40% B from 6.8–7 min before equilibration for 3 min with the initial conditions. The flow from the UPLC was coupled to a TimsTOF Flex (Bruker) instrument for mass spectrometric analysis, operated in both positive and negative ion modes. Compounds were annotated in Metaboscape (Bruker) using both an in-built rule-based annotation approach and using the Lipid-Blast MS2 library[105]. Features were removed if their average signal not were >5× more abundant in the QC samples than blanks (water extraction). The signals were normalized to internal standards in the SPLASH mix before correction for signal drift using the statTarget R package[106]. Finally, signals were normalized using the QC samples[107]. Peak intensities from each measured metabolite were scaled using Pareto scaling with *MetabolAnalyze* (version 1.3.1) and $\log_2$ transformed. We detected differentially abundant lipids using a linear model fitted separately for each lipid using limma[69], using the model formula *E = 0 + timepoint + timepoint:exposure* which estimates each lipid difference in abundance between control and lps samples from the same timepoint. We generated all statistical values using *ebayes*[69].

## Statistics and reproducibility

Study design is described in detail in "Animals" and "Exposure and dissection of maternal/fetal organs" sections. Sample size selection (*N* = 10 group/timepoint) was chosen based on (1) prior experience that six animals would be sufficient to detect moderate lung inflammation[108], (2) exclusion due to some male-only litters, and (3) consideration that downstream manifestation would be less pronounced than lung inflammation. Mice were not randomly distributed into experimental groups, as equal weights between groups were considered. The investigators were not blinded to allocation during experiments and outcome assessment. Overall reproducibility is shown in Fig. S1. For statistics of all data, see respective sections. Analyses were made using R 4.02 (https://www.r-project.org/) and visualizations using ggplot2[109] unless otherwise noted.

## Reporting summary

Further information on research design is available in the Nature Portfolio Reporting Summary linked to this article.

# Data availability

The RNA-seq data generated in this study have been deposited in the GEO database under accession code GSE224116. The proteomics raw data generated in this study have been deposited in the ProteomeXchange Consortium via the PRIDE partner repository under accession code PXD039402. The lipidomics raw data generated in this study have been deposited in the Metabolomics Workbench database under study ID code ST003125 [https://doi.org/10.21228/M8K43P]. Source data used for images are provided with this paper. The Lien et al. RNA-seq data[36] used in this study are available in the GEO database under accession code GSE151728. Source data are provided with this paper.

## Code availability

Code and parameters are available at https://github.com/signehansen/inflammation_to_metabolism[110].

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

## Acknowledgements

The authors would like to thank Navneet Vasistha, Diego García-González, and members of the Khodosevich lab (BRIC, University of Copenhagen), Noor Irman, Michael Guldbrandsen, Eva Terrida (National Research Centre for the Working Environment, Copenhagen, Denmark), Christian Vaagensø, Jette Bornholdt (Department of Biology and BRIC, University of Copenhagen), Histocore (BRIC and Finsen laboratory, University of Copenhagen) for help with experiments, computational analysis, discussion and ideas. This work was funded by grants from The Independent Research Fund Denmark (#7014-00120B to A.Sa. and K.S.H.), the Carlsberg Foundation (#CF19-0505 to A.Sa.) and the Novo Nordisk Foundation (#NNF20OC0059951 to A.Sa.). K.S.H. and U.V. were supported by Focused Research Effort on Chemicals in the Working Environment (FFIKA) from the Danish Government. S.S.K.H. was supported by Fru Birgit Levinsens grant. The lipidomics analyses were supported by the INTEGRA mass spectrometry research infrastructure for proteomics and metabolomics established at SDU by a generous grant from the Novo Nordisk Foundation (#NNF20OC0061575). B.B. was supported by Danish National Research Foundation (DNRF141, through ATLAS) and Independent Research Fund Denmark (#1026-00013B). The artwork in Fig. 1A and Fig. 10 were adapted from bioicons (https://bioicons.com/, CC 0 license). Other artworks included are originals.

## Author contributions

Conceptualization: A.Sa., K.S.H., and S.S.K.H. Supervision: A.Sa., K.S.H., K.K., and R.K. Experimental design: K.S.H., A.Sa., and S.S.K.H. Experimental work: S.S.K.H., A.P., A.C.T., J.R., K.S.H., J.H., N.J.F., V.A., B.B., B.E., and D.N. Data analysis: S.S.K.H., D.R., R.K., V.A., B.B., N.J.F., A.C.T., J.H., A.St., A.Sa., and K.S.H. Figure preparation: S.S.K.H. and A.Sa. First paper draft: S.S.K.H., A.Sa., and K.S.H. Writing and editing: S.S.K.H., A.Sa., K.S.H., R.K., and U.B.V.

## Competing interests

The authors declare no competing interests.
