## [Peer Review File · Nature Communications]

REVIEWER COMMENTS

Reviewer #1 (Remarks to the Author):

Hansen et al. describe acute response of maternal LPS exposure in lungs on maternal (lung and spleen) and fetal (placenta and liver) organs. While interesting in motive, the study is purely descriptive with limited new knowledge gained. Many of the outcomes are well established paradigms in the field (i.e. IL-6 being an important mediator).

Major issues:

The conclusion that the placenta selectively avoided an innate inflammatory immune response is likely rather the consequence of MIA model used not being robust enough. There are several studies that find differential responses in placenta and fetus after MIA (i.e. 36417858).

One of the most interesting findings was the metabolic dynamics but without maternal metabolic flux information it is impossible to assess whether simply an indirect effect of maternal metabolic response to LPS (a well-established consequence after acute LPS exposure) as opposed to the intriguing interpretation of an adaptive response initiated by the placenta as suggested in text.

There is only one validation of suggested pathway dynamics (ER stress in Figure 4G).

There will be different immune responses and developmental consequences triggered by LPS and PolyIC (administration site, dose, even PAMP batch will all contribute to how inflammatory any particular MIA outcome). The level of maternal cytokines known to mediate MIA is a critical measurement to assess how data fit into current literature. Currently only provide abundance change (not actual levels) of circulating maternal IL-6 and some other chemokines.

The sentence "How the placenta and the fetus respond to acute MIA over time is unknown." in the abstract is false and suggests a lack of understanding of the field.

I assumed there would be more integration of omic data than provided given the description in the abstract.

Sex should be considered given its importance in MIA outcomes.

Reviewer #2 (Remarks to the Author):

The authors have undertaken a tremendous amount of various whole-tissue-level omics in order to characterize the response over time of the maternal-fetal interface to maternal immune activation from inhaled LPS. This is of interest to the broad readership of this journal and to the field of reproductive immunology. The notion of fetal and placental sensing of the LPS in the form of metabolic adaptations is interesting. The work is original and has the potential to serve as a reference in the field of fetal outcomes following maternal immune activation. The authors do acknowledge that this is a descriptive study, and never overstate the conclusions drawn from their data. The authors show that:

- 1) Inhaled LPS causes expected proinflammatory gene expression in the lung, to a lesser extent in the maternal liver, and to a very small extent at the maternal-fetal interface and in the fetal liver. Based on prior published data, the maternal lung and placenta/decidua have overlapping and non-overlapping responses to direct exposure to LPS (inhaled vs intra-uterine).
- 2) IL6 protein is present in maternal serum post inhaled LPS, which correlates with IL6 transcript in the lung. Decidua and placenta express IL6 receptor components that may sense this circulating IL6.
- 3) In the decidua/placenta, genes associated with cell adhesion were induced early post-LPS, while ribosome biogenesis/protein synthesis and ER stress were induced at later timepoints.
- 4) Changes in the phosphoproteome were observed early post inhaled LPS. Proteins with changed phosphosites correlated with DEGs found in the RNAseq dataset early post inhaled LPS.
- 5) Fetal liver expressed genes associated with fatty acid and protein catabolism early post inhaled LPS, later resuming energy production by oxphos and cellular anabolism. This switch correlates with the period of increased ribosome activity in the placenta above.
- 6) DHA-containing lipids were increased in fetal liver post inhaled LPS

The following are more major suggestions. Overall, the major improvement I hope to see in the next iteration of this manuscript is a better setup of the scientific questions being asked, along with more integration of the many different omics modalities.

- 1) There is a large literature alluded to briefly and very generally in the introduction about the significance of MIA on fetal outcomes. What is the significance of this particular model of inhaled LPS? Are there prior data from the authors or from other groups about how inhaled LPS might impact the fetus? If there are specific fetal effects, the paper and analyses would have a greater focus, and the story would be clearer and higher impact.

2) In my opinion, the paper would benefit from analyses that revolve around downstream effects of a specific molecule(s), such as IL6. Doesn't have to be IL6, but I mention it because a main Figure (Fig. 2) is dedicated to finding the source of IL-6 and defining responsivity to IL-6 in the placenta. Do any of the other data support that IL6 plays a role in the genes, phospho-proteins, and lipids differentially evident post inhaled LPS? These do not need to be new bench experiments, rather textual changes and reanalysis of some of the omics data through the lens of IL6 or another molecule(s) of interest. There are recent data supporting a critical role for IL-6 in effects of MIA on the fetus (A. I. Lim et al., *Science* 373, eabf3002 (2021). DOI: 10.1126/science.abf3002). The Lim experimental system is much different (infection as a source of MIA, and IL6 acting on the fetal gut), but for example, perhaps IL6 impacts the fetal liver lipid changes seen in Figure 6?

3) The Figure 3 schematics illustrate this point. Can you help the reader understand the potential links among these biological processes? This could be a nice summative figure at the end of the manuscript.

4) Figure 3G is the only validation-style figure in the manuscript. The authors clearly explain the rationale for looking at ATF-4 in the placenta, but why was this the only and most important validation experiment to perform? Any chance ATF-4 links back to IL6 or another molecule of interest? If not, are there other targets of IL6 (or phospho-proteins downstream of IL6 signaling) that can be validated on already banked slides from these experiments?

5) Re the presentation of Fig 3G, please include a zoomed out view to pair with the zoomed in views. It is hard for me to get my bearings and identify the complex microvascular network that is characteristic of the placental labyrinth. At this magnification, I can clearly see induction of ATF-4 by IHC, but it is very difficult for the reader to assess subcellular localization by IHC at this magnification.

6) Figure 1E presents an opportunity to dive deeper into what makes the maternal lung LPS response different from the placental LPS response. Since this is primarily a bioinformatics paper, a more thorough analysis of these two responses would be of great interest.

7) Similar question for 1D. Are there enough data to conclude that these data represent distinct responses to LPS vs that the placenta is only seeing a minute amount of proinflammatory molecules? Are there other IL-6-stimulated genes that support the placenta seeing IL6 from the lung?

8) Given the many complex interactions possible in the data, I would expand the discussion substantially to again help the reader link all of the findings in a clearer way.

Minor comments are listed below:

1) In Figure 1A, please clarify in the picture and legend that this is intratracheal instillation. It is already clear in the main text but was confusing in the figure and legend. Please add gestational day 17 to the figure.

2) Gestational day 17 placentas have very slim deciduae, as they are almost fully involuted at this point. It is difficult to guarantee gross separation of these two layers, even with a dissecting microscope. I think this is reflected in the overall very similar gene expression profiles between decidua and placenta, and you acknowledge this on line 153, but I would state the difficulty explicitly. Please clarify that these are decidua-enriched samples and placenta-enriched samples.

- 3) Please label columns in Fig 1B. It was not immediately obvious that each column corresponded to the organ above. Why is only the bottom half of the figure annotated with gene families? Anything interesting and equally important in the top half?
- 4) Several sentences abruptly end with “consistent with” and a reference. Please indicate in words with which lines of evidence these data are consistent.
- 5) Line 138-141 What about type I interferon responsive genes? Is this known in the LPS lung literature? Might these genes account for important fetal outcomes? That seems to be the most profound change per figure S1H.
- 6) Line 143 and general question is: were maternal livers perfused prior to harvest? If not, some gene expression changes may be due to systemic uptake of LPS in the lung with changes in circulating intravascular leukocytes in the liver.
- 7) Line 217 says Figure 5A. Meant to say Figure 2A, correct?
- 8) Figure 5, can you please add the time labeling?
- 9) Line 538: genes downregulated at 5h? Looks like cellular respiration genes are in 5B, not 5C. Though neither B nor C appears to have any changes shown at 5 hours.
- 10) Related to Figure 6A: Agree that PG and PC are down at 24 hours. PCs are not labeled with 22:6 side chains as called out in text.
- 11) Line 607, some are not seen or are mislabeled in the figure. I cannot find PG 14:0_15:0. For PC 22:6_22:6, I see only PE 22:6_22:6. Please confirm the text call-outs and figure labeling.

Reviewer #3 (Remarks to the Author):

I was asked to review the phosphoproteomics section of this grant. The experimental design and execution were solid, and all files were available for reviewer examination. I have no recommendations for additional work. Likewise, the RNAseq and lipidomics work appear to be well-designed and executed.

I will yield to other reviewers regarding the immunology side of the study.

Reviewer #1 (Remarks to the Author):

Hansen et al. describe acute response of maternal LPS exposure in lungs on maternal (lung and spleen) and fetal (placenta and liver) organs. While interesting in motive, the study is purely descriptive with limited new knowledge gained. Many of the outcomes are well established paradigms in the field (i.e. IL-6 being an important mediator).

Reply: The novelty statement by the referee is disputed - we note that referee 2 states that "This is of interest to the broad readership of this journal and to the field of reproductive immunology.", "The work is original and has the potential to serve as a reference in the field of fetal outcomes following maternal immune activation" and "The authors do acknowledge that this is a descriptive study, and never overstate the conclusions drawn from their data": .

Note that we have not investigated maternal spleen: we have profiled maternal lung and liver, the placenta, and fetal liver. Importantly, to our knowledge, this study is the first to profile MIA following LPS lung exposure. Hence, we believe our study constitutes a significant first contribution to the study of LPS, using a route with much relevance to the human situation where gram negative bacterial infections in the airways are common during pregnancy. In the reworked Introduction and Discussion, we have tried to make this point clearer.

In the updated manuscript, we show evidence that LPS instillation of the lung leads to a higher upregulation of IL6 receptors in the decidua, and that SOCS3 is upregulated by LPS in the junctional zone of the placenta: this is important as SOCS3 is a key repressor of IL6 signalling (see figures 2-3).

Major issues:

1: The conclusion that the placenta selectively avoided an innate inflammatory immune response is likely rather the consequence of MIA model used not being robust enough. There are several studies that find differential responses in placenta and fetus after MIA (i.e. 36417858).

Reply: Thank you for addressing this important point (that to some degree was also touched upon by reviewer 2). It is true that several studies observe differential responses in placenta and fetus after MIA, among those the paper highlighted by the referee 36417858. We have not been clear enough on this point.

An important difference between ours and many other studies, in particular the study referred to above, is that in our study, we instill LPS in the lung of the mother rather than systemic exposure by injection (for example, the study referred to above uses intra-peritoneal injection of pIC). To our knowledge our study is the first to characterise placental and fetal LPS response following indirect LPS exposure through the lungs.

These modes of MIA are different, and we address this point explicitly in the paper: the placenta can react to direct LPS exposure by upregulating innate immune response genes, but this does not occur in our study setting with LPS exposure occurs in the lung (See figure 1E, D), with our LPS dose.

That said, we agree that the extent of the innate inflammatory immune response will depend on the amount of administered inflammogen, in our case LPS. In this study, we aimed for a physiologically relevant LPS dose mimicking a substantial lung inflammation, but with limited maternal toxicity, designed so that preterm birth was not induced - since the latter constitutes an extreme scenario. With our LPS dose, the RNA-seq results from lung is certainly very robust, with strong inflammatory responses at RNA level and decreased maternal weight gain (by approximately 5%) and increased neutrophil influx in bronchoalveolar fluid lasting for 24h in lung of non-pregnant animals (Fig.S1.B,). Furthermore, the maternal liver mounted a clear and robust innate immune response. Taken together, these observations are consistent with LPS inducing strong inflammation in the maternal organism.

Therefore, we believe that our study provides valuable insights that contribute to the existing literature on MIA. We have now added a section in the introduction specifying our focus on airway LPS exposure, and in the discussion we acknowledge that other responses might be observed at higher LPS dose levels or other routes of exposure.

2. One of the most interesting findings was the metabolic dynamics but without maternal metabolic flux information it is impossible to assess whether simply an indirect effect of maternal metabolic response to LPS (a well-established consequence after acute LPS exposure) as opposed to the intriguing interpretation of an adaptive response initiated by the placenta as suggested in text.

Reply: Thanks for this point and noting its interest. In the original manuscript, we did discuss both hypothesis: DHA may be incorporated from local deposits in the fetal liver or be transferred from the mother within the time frame of the study.

In the current video of the manuscript, we performed lipidomics on maternal liver and plasma at all time points, and then made an integrative analysis of DHA chains and precursors (Fig.9). This showed that, as the referee pointed out, in the maternal liver, DHA chains and lipids with incorporated DHA are induced by LPS at 5-24h. Interestingly, the abundance of chains showed a time shift between maternal liver and fetal liver which is compatible with transfer of such chains through the placenta during the time period studied. So, this hypothesis is supported by the data.

That said, the notion that the fetal liver uses its own deposits to incorporate DHA into larger lipids also has merits, as we observe RNA expression changes of many of the genes responsible for such conversions as early as 5h (transfer from the mother takes, at minimum, several hours, which is not compatible with the early RNA expression of these genes). Thus, it is likely that both processes are active, and we now acknowledge in the Discussion that to characterise the importance of each path, flux- or labelling approaches will be necessary.

3: There is only one validation of suggested pathway dynamics (ER stress in Figure 4G).

Reply: We now present additional validations, including immunohistochemistry data on SOCS3 (Figure 3), and additional lipidomics data from maternal liver and maternal plasma.. We would like to point out that in many cases (e.g. in fetal and maternal liver), the dynamics identified by RNA-seq were validated by lipidomics data.

4: There will be different immune responses and developmental consequences triggered by LPS and PolyIC (administration site, dose, even PAMP batch will all contribute to how inflammatory any particular MIA outcome).

Reply: Thanks for pointing out this important issue. Several studies have documented that the specific PAMP and administration site hugely influences the MIA outcome, albeit also many common pathways are triggered, especially at the port of entrance. This is why it is so important to characterise each of the relevant routes of exposure for each of the common PAMPs (and at a range of dose levels) - only in this way can we together build up a database on MIA with the common goal to aid delineation of the potential resolve of the sequelae of MIA in humans. We therefore agree that it is important to acknowledge that the responses we

describe are not universal for any direct and indirect MIA. We now address this in the discussion, in the limitations section.

5: The level of maternal cytokines known to mediate MIA is a critical measurement to assess how data fit into current literature. Currently only provide abundance change (not actual levels) of circulating maternal IL-6 and some other chemokines.

Reply: In the current version of Figure 2, panels A-D show abundance changes because that was the most relevant statistic when comparing cytokine concentration changes with expression changes. We agree that absolute levels are also relevant: we now also plotted average concentrations (pg/mL) of all measured cytokines in plasma in Fig. S2A.

6: The sentence “How the placenta and the fetus respond to acute MIA over time is unknown.” in the abstract is false and suggests a lack of understanding of the field.

Reply: Please accept our excuses for not being clearer. What we meant to say was that the response to *indirect* MIA originating from exposure to LPS via the airways (i.e. indirect MIA) is not well described.

We also believe that the time series response across many organs following indirect MIA have only been done to a very limited extent previously, especially considering the extensive omics-based characterisation in the present study. Accordingly, the abstract sentence has been modified to:

“Because it is not well understood how the placenta and fetus respond to acute pulmonary inflammation, we characterized the response to pulmonary LPS exposure across 24h in maternal and fetal organs using multi-omics, imaging and integrative analyses”

7: I assumed there would be more integration of omic data than provided given the description in the abstract.

We are somewhat surprised by this comment as in the previous version of the manuscript, 12 figure panels in 4 main and 2 supplementary figure showed one or more integrative analyses (we interpret this as analyses that compare two or more data modalities or studies) - figure indices below are updated to the new manuscript:

- Figure 1 D,E show RNA expression change comparison between two different studies and three tissues

- Figures 2A,B, C, D and S2 B,C show comparisons between cytokine change in blood vs RNA expression in tissues
- Figure 6B shows comparisons between changes in protein phosphorylation and RNA expression,
- Figure 8C show comparison between lipid abundance change and gene expression abundance

We have now complemented the above with;

- Figure S1 K analyzes differences in GO terms from direct LPS exposure in placenta and maternal lung from two studies.
- Figure 2G compares Socs3 gene expression in two tissues with cytokine concentrations.
- Fig S6A and B now show results from a DIABLO discriminant analysis (from the mixOmics R package) which identifies multi-omic signatures that separates LPS/Ctrl samples based on an integration of lipidomics and RNA-seq data. The patterns observed here largely agrees with the patterns that we found by analyzing each set individually.
- Fig S7 A, B shows the same type of analysis for maternal liver,

We made attempts at using the MixOmics modelling framework to integrate RNAseq and phospho-proteomics data, but the results were hard to interpret - perhaps because the functional implications of changes in phospho-sites are generally hard to interpret - so we left this out of the manuscript.

7. Sex should be considered given its importance in MIA outcomes.

Reply: We agree that sex is an important parameter for the outcomes of maternal immune activation (MIA) on the placenta and fetus: it is well established that there is pronounced sexual dimorphism in immunity and accumulating evidence demonstrates that this dimorphism translates into also dimorphic phenotypic responses in the placenta as well as the fetus.

Our experiment was designed accordingly: we only assessed female mice, including fetuses. For the latter, for distinguishing sex, we did not rely solely on anatomy but also used genotyping, as described in Methods.

While we recognize the significance of sex as a variable, we believe that our focus on female fetuses provides valuable insights that contribute to the existing literature on MIA. We now

address this in the manuscript discussion that the exclusive focus on female fetuses presents a limitation and that it is important to include both sexes in future studies.

Reviewer #2 (Remarks to the Author):

The authors have undertaken a tremendous amount of various whole-tissue-level omics in order to characterize the response over time of the maternal-fetal interface to maternal immune activation from inhaled LPS. This is of interest to the broad readership of this journal and to the field of reproductive immunology. The notion of fetal and placental sensing of the LPS in the form of metabolic adaptations is interesting. The work is original and has the potential to serve as a reference in the field of fetal outcomes following maternal immune activation. The authors do acknowledge that this is a descriptive study, and never overstate the conclusions drawn from their data. The authors show that:

1) Inhaled LPS causes expected proinflammatory gene expression in the lung, to a lesser extent in the maternal liver, and to a very small extent at the maternal-fetal interface and in the fetal liver. Based on prior published data, the maternal lung and placenta/decidua have overlapping and non-overlapping responses to direct exposure to LPS (inhaled vs intra-uterine).

2) IL6 protein is present in maternal serum post inhaled LPS, which correlates with IL6 transcript in the lung. Decidua and placenta express IL6 receptor components that may sense this circulating IL6.

3) In the decidua/placenta, genes associated with cell adhesion were induced early post-LPS, while ribosome biogenesis/protein synthesis and ER stress were induced at later timepoints.

4) Changes in the phosphoproteome were observed early post inhaled LPS. Proteins with changed phosphosites correlated with DEGs found in the RNAseq dataset early post inhaled LPS.

5) Fetal liver expressed genes associated with fatty acid and protein catabolism early post inhaled LPS, later resuming energy production by oxphos and cellular anabolism. This switch correlates with the period of increased ribosome activity in the placenta above.

6) DHA-containing lipids were increased in fetal liver post inhaled LPS

Reply: Thank you very much for this positive and very constructive review that helped us improve the manuscript, and excellent summary above.

The following are more major suggestions. Overall, the major improvement I hope to see in the next iteration of this manuscript is a better setup of the scientific questions being asked, along with more integration of the many different omics modalities.

1) There is a large literature alluded to briefly and very generally in the introduction about the significance of MIA on fetal outcome prior data from the authors or from other groups about how inhaled LPS might impact the fetus? If there are specific fetal effects, the paper and analyses would have a greater focus, and the story would be clearer and higher impact.

Reply: Thank you for this great point.

We agree that the story might have been clearer if specific fetal effects had already been reported and we could have based our hypothesis on these. However, we have searched repeatedly for studies of gestational inhalation of LPS but have so far failed to identify experimental studies on how inhaled LPS might impact the fetus, from our own and the hands of other groups. This is illustrated well by a comprehensive systematic review of maternal immune activation in rodents and neurodevelopmental changes in gene expression and epigenetic modulation in offspring brain encompassed 118 studies, of which none used the pulmonary route for administration of LPS (Woods, R. M. *et al. Neurosci. Biobehav. Rev.* **129**, 389–421 (2021)). It is therefore surprising that LPS has not been studied in this context, as the Gram-negative bacteria of *Haemophilus influenzae* is the second most common cause of community acquired pneumonia, with also other Gram negative bacteria contributing to this disease. In contrast, only about 10% of the community acquired pneumonia are ascribed to viral infections, albeit this number might be higher, especially if co-exposure with bacteria is taken into consideration (Shoar and Musher, *Pneumonia* 2020 Oct 5:12:11. doi: 10.1186/s41479-020-00074-3. eCollection 2020.) Of note, inhalation of dust containing dead or live Gram-negative bacteria, in occupational and home settings, might also contribute considerably to airway inflammatory load. The present study therefore constitutes a significant first contribution to the study of LPS, using a route with much relevance from the human situation.

We have now incorporated part of this response in the introduction and revisit this important point in discussion.

2) In my opinion, the paper would benefit from analyses that revolve around downstream effects of a specific molecule(s), such as IL6. Doesn't have to be IL6, but I mention it because a main Figure (Fig. 2) is dedicated to finding the source of IL-6 and defining responsivity to IL-6 in the placenta. Do any of the other data support that IL6 plays a role in the genes, phospho-proteins, and lipids differentially evident post inhaled LPS? These do not need to be new bench experiments, rather textual changes

and reanalysis of some of the omics data through the lens of IL6 or another molecule(s) of interest. There are recent data supporting a critical role for IL-6 in effects of MIA on the fetus (A. I. Lim et al., Science 373, eabf3002 (2021). DOI: 10.1126/science.abf3002). The Lim experimental system is much different (infection as a source of MIA, and IL6 acting on the fetal gut), but for example, perhaps IL6 impacts the fetal liver lipid changes seen in Figure 6?

Reply: Thanks for this excellent point. Based on new exploratory analysis of such IL6 downstream effects, we show that SOCS3, which is a part of the IL6 pathway but also is the key repressor of IL6 signalling, is upregulated by LPS in the placenta but less so in decidua, and that the placenta-specific upregulation is correlated to the levels of IL6 in the blood (Fig 2G). We validated this on protein level using ICH (Fig 3), which showed that there also was an increase in SOCS3 protein expression in LPS mice which was particularly intense in spongiotrophoblasts of the junctional zone, that serves as a separator of maternal decidua and the labyrinth zone, and as structural support of the latter. Overexpression of SOCS3 in the junctional zone could thus partially protect the labyrinth from induction of the IL-6-Jak/STAT-pathway.

3) The Figure 3 schematics illustrate this point. Can you help the reader understand the potential links among these biological processes? This could be a nice summative figure at the end of the manuscript.

Reply: Thanks for this excellent point. We have now added a new Fig 10 as a summary figure that we use to improve the discussion section: it summarises our main observations and our interpretations of them and discusses links between processes.

4) Figure 3G is the only validation-style figure in the manuscript. The authors clearly explain the rationale for looking at ATF-4 in the placenta, but why was this the only and most important validation experiment to perform? Any chance ATF-4 links back to IL6 or another molecule of interest? If not, are there other targets of IL6 (or phospho-proteins downstream of IL6 signaling) that can be validated on already banked slides from these experiments?

Reply: We were interested in ATF-4 because we thought that the ER stress that the GO analysis identified was interesting. Three ER-stress pathways exist, and they are sequentially activated upon increasingly severe ER stress - we started with ATF-4 because its translocation

to the nucleus indicates activation of the 'mildest' ER stress pathway. We tried staining for other ER stress markers all associated with more severe ER stress and none of these worked out due to technical challenges and/or no substantial change LPS vs Ctrl. We believe that the activation of mild ER stress corresponds well to the mRNA results.

As to other IL6 targets, see the reply to point 2: SOCS3 is a downstream target of IL6 but also is a repressor of the pathway, and was overexpressed in the placenta as a function of IL6 concentration. As noted above, overexpression of SOCS3 could repress the induction of the IL-6-Jak/STAT-pathway and thus an immune response in the placenta.

5) Re the presentation of Fig 3G, please include a zoomed out view to pair with the zoomed in views. It is hard for me to get my bearings and identify the complex microvascular network that is characteristic of the placental labyrinth. At this magnification, I can clearly see induction of ATF-4 by IHC, but it is very difficult for the reader to assess subcellular localization by IHC at this magnification.

Reply: This has now been done. We remade the IHC and now also include different zoom levels: see Fig. 5.

6) Figure 1E presents an opportunity to dive deeper into what makes the maternal lung LPS response different from the placental LPS response. Since this is primarily a bioinformatics paper, a more thorough analysis of these two responses would be of great interest.

Reply: We interpret this as a request to compare the direct LPS response of placenta (from Lien, Y.-C. et al.. Front. Physiol. 11, 592689 (2020)) to that of lung (our study). We initially did not pursue this in depth since the analysis is based on comparing data two different studies with somewhat different experimental setups. However, in our opinion, what Fig1E shows is how remarkably similar these responses are - there are not many genes that stand out as being up-or down regulated in only one tissue. To address this request, we have now made GO analyses based on ranking gene by their LPS vs Ctrl log₂FC in each tissue: this is now shown in Figure S1K. Briefly, the GO analysis highlights the similarity rather than the differences in response - the only real difference is that upregulated genes in the placenta were enriched for genes associated with regulation of immune response, which was not the case for the lung.

7) Similar question for 1D.

Are there enough data to conclude that these data represent distinct responses to LPS vs that the placenta is only seeing a minute amount of proinflammatory molecules? Are there other IL-6-stimulated genes that support the placenta seeing IL6 from the lung?

Reply: Given that this is a mostly descriptive, genomics-based study, we cannot conclusively prove this. However, we believe that it is a distinct response based by the following observations:

Since the maternal liver reacts strongly to LPS in the lung, it is hard to argue that there is minute amount of proinflammatory molecules as such in the blood stream (as requested by referee 1, we now show a new supplementary figure with actual cytokine concentrations in the blood - Fig S2A) , and one would expect that the placenta would 'see' these proinflammatory molecules as well as it has access to the same blood stream (and, notably, it has expressed receptors for e.g. IL6 that is clearly induced in the blood stream, as discussed above).

As described above, we have reworked Figure 2 two to focus on Il6 downstream signalling where the most interesting finding was that Socs3 is specifically upregulated in the placenta as a response to LPS instillation in the lungs. Socs3 is regulated by the STAT/Il6 signalling pathway but subsequently represses Il6 signalling, which would fit with our results that the placenta does not upregulate innate immune response genes like the maternal and fetal lung do.

8) Given the many complex interactions possible in the data, I would expand the discussion substantially to again help the reader link all of the findings in a clearer way.

Reply: This has now been done: as requested by referee1 we have made a stronger point on our setup (LPS in lungs), added a new figure that summarises our findings and interpretations (Fig 10) and now also discuss our new lipidomics and SOCS3 data.

Minor comments are listed below:

1) In Figure 1A, please clarify in the picture and legend that this is intratracheal instillation. It is already clear in the main text but was confusing in the figure and legend. Please add gestational day 17 to the figure.

Reply: We have now added that the instillation is intratracheal and that gestation day.

2) Gestational day 17 placentas have very slim deciduae, as they are almost fully involuted at this point. It is difficult to guarantee gross separation of these two layers, even with a dissecting microscope. I think this is reflected in the overall very similar gene expression profiles between decidua and placenta, and you acknowledge this on line 153, but I would state the difficulty explicitly. Please clarify that these are decidua-enriched samples and placenta-enriched samples.

Reply: Thank you is an excellent point. We have included this, but it made the most sense to do this already at the start of results where we describe the samples, and then revisit this when we discuss the similarity of expression response patterns. Do note however, that the actual gene expression patterns between these two samples are different (see e.g. the PCA in Figure S1E) , but the *response* to LPS is overall similar for most genes.

3) Please label columns in Fig 1B. It was not immediately obvious that each column corresponded to the organ above. Why is only the bottom half of the figure annotated with gene families? Anything interesting and equally important in the top half?

Reply: We have now labelled the columns in Fig 1B.

The reason for only annotating GO terms, gene families etc for the lower clusters is that

i) it dominates the clustering and

ii) it is the clusters that are the most interpretable, and also illustrates the large immune reaction in maternal lung that is less prevalent in maternal liver, and then largely not present in the placenta and fetal liver.

Note that GO term enrichment analysis and their interpretation for placenta and fetal liver are done in depth in Figures 2 and 7

4) Several sentences abruptly end with “consistent with” and a reference. Please indicate in words with which lines of evidence these data are consistent.

Reply: Thanks for this point: we were challenged by the word count limit so many of these were cut short. We have now extended many of these where we thought it was the most relevant.

5) Line 138-141 What about type I interferon responsive genes? Is this known in the LPS lung literature? Might these genes account for important fetal outcomes? That seems to be the most profound change per figure S1H.

Reply: Type I interferon genes are indeed important regulators of lung inflammation (e.g. see this review: Makris et al, [10.3389/fimmu.2017.00259](https://doi.org/10.3389/fimmu.2017.00259)). However, this statement is also true for genes associated with many of the other highly enriched GO terms in lung. We believe that given our data, it is too early to speculate about the roles that a specific group of genes that are generally induced by lung inflammation play in fetal outcomes.

6) Line 143 and general question is: were maternal livers perfused prior to harvest? If not, some gene expression changes may be due to systemic uptake of LPS in the lung with changes in circulating intravascular leukocytes in the liver.

Reply: Maternal livers were not perfused. In general, observed gene expression differences in tissues may be due to

- i) gene expression differences in cells that are present in the tissue throughout the time course,
- ii) altered cell composition, for instance leukocytes or immune cell uptake or otherwise altered blood composition
- iii) or some combination of the i and ii.

This cannot be distinguished using bulk RNA-seq analysis.

That said, if there were large changes in circulating cells in the bloodstream and this would contribute to large changes in liver expression, we would also expect to see the same pattern in placenta, which we do not. The same is likely true for LPS being carried by the blood stream: here, Figure 1D,E clearly shows that the placentas in our study do not respond like placentas that are directly exposed to LPS.

7) Line 217 says Figure 5A. Meant to say Figure 2A, correct?

Reply: Many thanks for spotting this: this is now corrected.

8) Figure 5, can you please add the time labeling?

Reply: Many thanks for spotting this - we assume this refers to the missing X axis labelling on the heat maps: this is now corrected.

9) Line 538: genes downregulated at 5h? Looks like cellular respiration genes are in 5B, not 5C. Though neither B nor C appears to have any changes shown at 5 hours.

Reply: Thanks for finding this typo: it is indeed Fig. 5b (figure 5 is Fig 7 in the new manuscript) and at 2h, not 5h. This is now fixed.

10) Related to Figure 6A: Agree that PG and PC are down at 24 hours. PCs are not labeled with 22:6 side chains as called out in text.

Reply: we now label all lipids having one or more 22:6 chains with bold - this includes not only the PC class.

11) Line 607, some are not seen or are mislabeled in the figure. I cannot find PG 14:0_15:0. For PC 22:6_22:6, I see only PE 22:6_22:6. Please confirm the text call-outs and figure labeling.

Reply: These lipids were shown in the supplement, but we agree that was confusing: we have reworked the network figure to now show the lipids we discuss in main text, and show their changes over time in the main figure.

Reviewer #3 (Remarks to the Author):

I was asked to review the phosphoproteomics section of this grant. The experimental design and execution were solid, and all files were available for reviewer examination. I have no recommendations for additional work. Likewise, the RNAseq and lipidomics work appear to be well-designed and executed.

I will yield to other reviewers regarding the immunology side of the study.

Reply: Many thanks for the review.

EVIEWERS' COMMENTS

Reviewer #1 (Remarks to the Author):

The majority of my comments were addressed.

Reviewer #2 (Remarks to the Author):

In my opinion, the authors took feedback seriously and have improved the manuscript's readability and cohesiveness dramatically. I like the addition of the junctional zone SOCS3 angle of the story to provide mechanistic evidence of why the placenta experiences a uniquely dampened response to IL-6 in this model. The summative Figure 10 is terrific. The histology findings in Figures 3 and 5 are presented clearly and convincingly.

Tiny textual change required prior to publication:

My version of the manuscript has this in the first paragraph of the discussion: "[small sentence on why we do it and that this is distinct from previous stuff that mostly did not look at lung administration - rephrase some stuff from introduction]."

Reviewer #2 (Remarks on code availability):

I am not qualified to assess the soundness of their code, as I am an amateur coder at best. A bioinformatics expert would be needed for a rigorous assessment of this.

Response to referee comments

Reviewer #1 (Remarks to the Author):

The majority of my comments were addressed.

Reply: No reply necessary

Reviewer #2 (Remarks to the Author):

In my opinion, the authors took feedback seriously and have improved the manuscript's readability and cohesiveness dramatically. I like the addition of the junctional zone SOCS3 angle of the story to provide mechanistic evidence of why the placenta experiences a uniquely dampened response to IL-6 in this model. The summative Figure 10 is terrific. The histology findings in Figures 3 and 5 are presented clearly and convincingly.

Tiny textual change required prior to publication:

My version of the manuscript has this in the first paragraph of the discussion: "[small sentence on why we do it and that this is distinct from previous stuff that mostly did not look at lung administration - rephrase some stuff from introduction]."

Reply: Thanks for noticing this comment that was accidentally left in the manuscript : it has now been deleted.

Reviewer #2 (Remarks on code availability):

I am not qualified to assess the soundness of their code, as I am an amateur coder at best. A bioinformatics expert would be needed for a rigorous assessment of this.

Reply: No reply necessary